# Next-generation biomonitoring of the early-life chemical exposome in neonatal and infant development

Thomas Jamnik[1], Mira Flasch [1], Dominik Braun [1], Yasmin Fareed [1], Daniel Wasinger[1], David Seki[2,3], David Berry [3], Angelika Berger[2], Lukas Wisgrill [2,4] & Benedikt Warth [1,4]✉

Exposure to synthetic and natural chemicals is a major environmental risk factor in the etiology of many chronic diseases. Investigating complex co-exposures is necessary for a holistic assessment in exposome-wide association studies. In this work, a sensitive liquid chromatography-tandem mass spectrometry approach was developed and validated. The assay enables the analysis of more than 80 highly-diverse xenobiotics in urine, serum/ plasma, and breast milk; with detection limits generally in the pg-ng mL$^{-1}$ range. In plasma of extremely-premature infants, 27 xenobiotics are identified; including contamination with plasticizers, perfluorinated alkylated substances and parabens. In breast milk samples collected longitudinally over the first 211 days post-partum, 29 analytes are detected, including pyrrolizidine- and tropane alkaloids which have not been identified in this matrix before. A preliminary estimation of daily toxicant intake via breast milk is conducted. In conclusion, we observe significant early-life co-exposure to multiple toxicants, and demonstrate the method's applicability for large-scale exposomics-type cohort studies.

[1] University of Vienna, Faculty of Chemistry, Department of Food Chemistry and Toxicology, Währinger Straße 38, 1090 Vienna, Austria. [2] Medical University of Vienna, Division of Neonatology, Pediatric Intensive Care and Neuropediatrics, Comprehensive Center for Pediatrics, Währinger Gürtel 18-20, 1090 Vienna, Austria. [3] Centre for Microbiology and Environmental Systems Science, Department of Microbiology and Ecosystem Science, Division of Microbial Ecology, University of Vienna, 1090 Vienna, Austria. [4] Exposome Austria, Research Infrastructure and National EIRENE Hub, Vienna, Austria. ✉email: benedikt.warth@univie.ac.at

The complex processes of disease etiology are typically driven by both, individual genetic predispositions and the environment. Mounting evidence suggests that environmental causes are predominant risk factors in clinical conditions, such as ischemic heart disease, lung disease and most cancers[1]. Complementing genomic research, the study of how genetic changes and predispositions affect gene product functions which may lead to disease, by taking the holistic environmental influences into account is key to bridging the gap between the knowledge of underlying disease mechanisms and epidemiology[2–4]. The exposome paradigm was first introduced by Wild in 2005, and is defined as the multitude of exposures an individual experiences throughout a lifetime[5]. This encompasses an exceptional variety of dynamic external factors that include environmental-, diet-, social- and lifestyle influences, and associated biological responses; all of which impart specific fingerprints on our internal environment[6]. Exposure to certain man-made and naturally occurring chemicals has long been recognized as a prime risk factor. These xenobiotics have the potential to enter an organism through diet, water, air, cosmetics, personal-care products, plastics, anti-microbial products and many other sources[7]. However, while genome-wide association studies (GWAS) were enabled by technical advancements in the past, the required tools to establish an environmental counterpart are still missing to date[2].

The Developmental Origins of Health and Disease (DOHaD) hypothesis has postulated that exposure to certain environmental influences during critical periods of development and growth may have significant negative long-term health effects[8]. Recently, special attention has been devoted to the assessment and characterization of endocrine disruptors during early life, as evidence has confirmed the DOHaD hypothesis concerning exposure to this group of chemicals[9–11].

Currently available human biomonitoring (HBM) data and methods, however, clearly fall short of the need for holistic exposome research. Contrary to GWAS, the assessment of chemical exposure has historically primarily relied on self-reported questionnaires concerning food consumption and product usage[12]. Particularly related to quantity, data that is dependent on personal memory (i.e. 24 h dietary recalls) is unreliable, while duplicate diet studies are time and resource-intensive. Furthermore, mere environmental and food monitoring (i.e. the analysis of food products and environmental pollution) does not account for inter-individual variation in exposure, absorption and xenobiotic metabolism. Moreover, toxicological and health-based guidance values are usually derived from animal or in vitro experiments that rely on the individual toxicity of the assessed agents, rather than considering co-exposure and mixture effects[13].

Mostly single chemicals or chemical classes have been examined in large HBM cohort studies, while in toxicological research the "cocktail effects" caused by xenobiotics from different sources are still largely neglected[14–17]. As it is more straightforward to address the adverse consequences of our modern industrialized lifestyle rather than assessing exposure to environmental chemicals over millennia, little attention has been given to naturally occurring chemicals compared to artificial compounds. Nonetheless, they also contribute to the overall toxic burden of an individual and can be responsible for adverse combinatorial effects. This highlights the need for holistic and simultaneous assessment of both, naturally occurring and synthetic chemicals. One of the few exposome-wide association studies (ExWAS) that have been undertaken, depended on questionnaires and multiple methods to assess different classes of chemical contamination (Agier et al. 2020[10]). As there is no analytical method to date that can cover several classes with sufficient sensitivity, a cost-, time- and sample-intensive approach had to be applied to obtain coverage of a broad range of chemicals.

The assessment of a multitude of organic xenobiotics at ultra-trace levels (<10 ng mL$^{-1}$) in exposome research can be achieved by liquid chromatography coupled to mass spectrometry (LC-MS). Recent advances in analytical instrumentation have expedited the detection of trace levels of xenobiotics in human tissues and bio-fluids. Thus, an improved quantitative assessment of the chemical burden of an individual is now feasible. While non-targeted, high-resolution methods are proposed as the future key technology for exposomic research; currently, targeted methods employing triple quadrupole instrumentation still provide higher sensitivity, linearity and reduced data processing needs[18]. The correlation of comprehensive biomonitoring data at a population level with epidemiological data provides an approach to conceptualize and unravel the contribution of chemicals to the development of disease and may enable individualized treatments for highly exposed patients in the context of precision medicine. Moreover, such information can support informed policy-making to protect, in particular, susceptible demographics from adverse chemical exposures via precision prevention (Fig. 1)[19] to, e.g. endocrine-disrupting phthalates[20] and toxic perfluorinated alkylated substances (PFAS)[21,22].

Holistic, simultaneous and sensitive monitoring of multiple chemical exposures is not yet available, therefore an ultra-sensitive LC-MS/MS method to determine a wide variety of synthetic and natural xenobiotics (>80) in three clinically relevant human bio-fluids (urine, serum/plasma and breast milk) is developed in this work. To demonstrate the potential of the method for detecting trace-level chemical contamination in precious, low-volume samples (<100 μL), obtained from a cohort of extremely premature babies are investigated. This study reports an unprecedented assessment of early-life exposure in the blood of a particularly vulnerable patient subgroup. Additionally, longitudinal breast milk samples covering the first 211 days of life from one mother are screened for potentially adverse quantities of xenobiotics and estimations of daily intake are related to current exposure reference values. We show that preterm neonates are exposed to 27 of the included compounds, partly to significantly higher levels than adult controls. Moreover, breast milk acts as a vehicle of 29 xenobiotics to the infant, however, not at critical levels according to current health recommendations. Most importantly, the method demonstrates high sensitivity and selectivity for multiple chemical classes, as well as an extendable nature. This enables its future application in large-scale cohorts.

## Results

**Compound coverage.** A total of 81 compounds from highly diverse chemical classes encompassing different structures, exposure levels and toxicological modes of action were selected. The compound library from a less sensitive and broad method only targeting endocrine-disrupting chemicals (EDCs) developed by Preind et al. (2019)[23] was used as a foundation and subsequently expanded with the addition of genotoxic compounds, carcinogens, neurotoxins and other markers of exposure from various sources. Importantly, several human biotransformation products were also included to account for human phase I and phase-II metabolism. The choice of chemicals was based on the toxicological potential (i.e. is the compound of relevance in a health context?), real-life exposure levels (i.e. does this compound occur in quantities feasible to be determined?), suitability of the instrumental platform (i.e. is retention via reversed-phase LC and ionization via ESI-MS sufficient for sensitive analysis?) and representativeness for each class of toxicants to guarantee thorough monitoring of multiple adverse exposures (i.e. does this compound usually occur alongside other relevant chemicals of the same chemical class?). The general sample preparation and

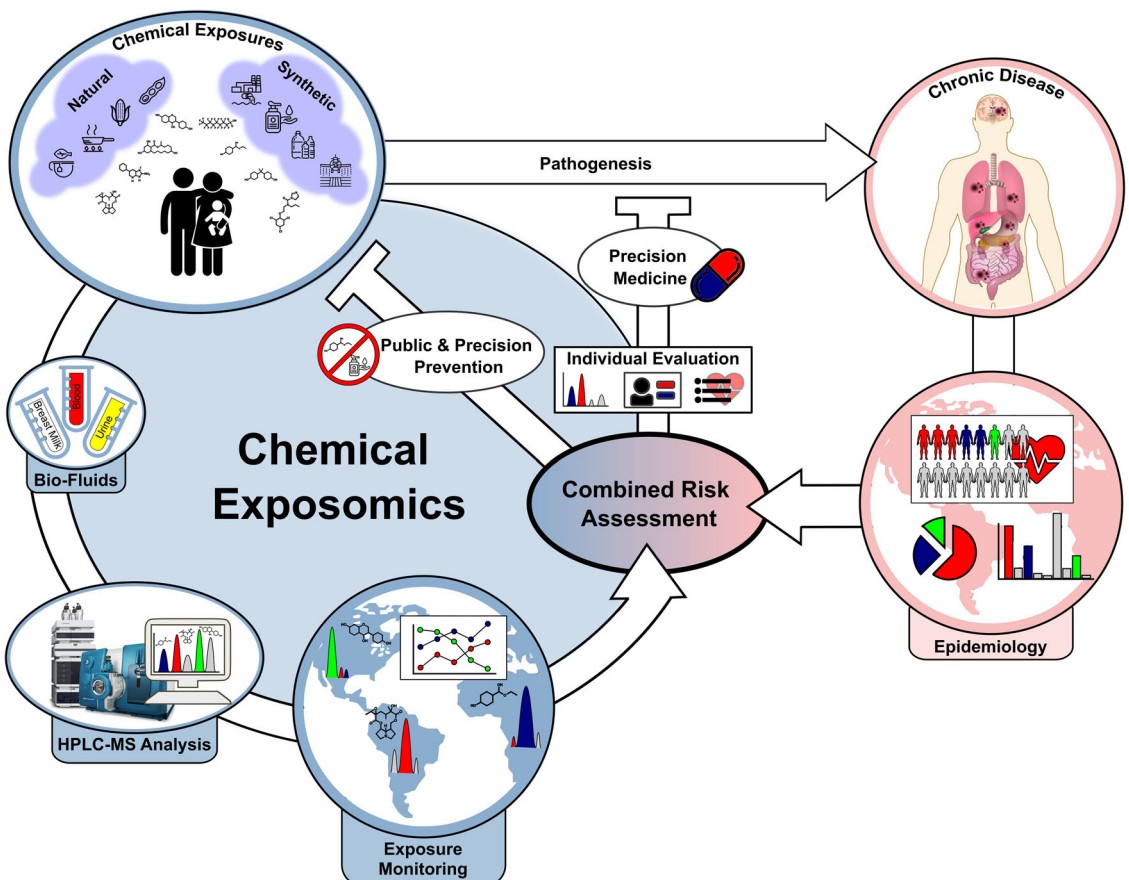

**Fig. 1 Comprehensive exposomic biomonitoring and combined risk assessment.** Schematic illustration depicting the application of a comprehensive exposome approach in an epidemiological setting to improve our understanding of the impact of chemical co-exposure on chronic disease. The prevention of disease will be aided by informed policy decisions and individualized precision medicine.

LC-MS workflow are depicted in Fig. 2 a, b. The chemical classes included in the assay and the number of compounds are shown in Fig. 2c. For information on the chemicals and reagents that were used in this study, refer to the Supplementary Information and Supplementary Table 1.

In addition to EDCs, the focus was directed toward the genotoxic and carcinogenic aristolochic acids, pyrrolizidine alkaloids and neurotoxic tropane alkaloids. These secondary plant metabolites have been known to contaminate agricultural crops, herbal tea remedies and plant food supplements. Notable concern has been raised with such plant metabolites because herbal teas are often specifically marketed toward pregnant- and lactating women[24]. The implication is that potential exposure of infants in utero and via breast milk is thus increased. Nephropathy due to chronic poisoning with aristolochic acids has long been demonstrated and studied[25]. Despite a high toxic potential, pyrrolizidine and tropane alkaloids are contaminants that have received little attention in the past[26,27]; and long-term exposure effects on humans have not yet been investigated. In addition, food-processing- and disinfection by-products that are formed during heating of foods rich in carbohydrates or protein, and during chlorination of water, respectively, were included. The former is known to have mutagenic activity, while the latter is increasingly associated with carcinogenicity and other non-carcinogenic toxic effects[28]. Finally, markers of smoke exposure were also included, e.g. metabolites of polycyclic aromatic hydrocarbons (PAHs) and nicotine.

**Implementation of comprehensive and ultra-sensitive LC-MS.**
The multitude of structural and physicochemical properties of

toxicants pose a major challenge for the concurrent analysis of many different chemical classes. For applicability in future, high-throughput exposome-scale studies, the sample extraction procedure must be comprehensive, yet still rapid. To deplete matrix components from 200 μL urine and serum, a one-phase extraction followed by protein precipitation was used. Conversely, a two-phase extraction was utilized for breast milk. Due to the diverse chemical properties of the extracted compounds, the chromatographic column had to retain molecules with a broad range of polarities, and also effectively separate several isomers with identical mass to charge (m/z) ratios. In addition to implementing a state-of-the-art UHPLC system, careful optimization of the chromatographic conditions including gradient, eluents, and the addition of $NH_4F$ as an organic modifier was essential. To maximize sensitivity, the ion optic parameters of the triple quadrupole mass spectrometer were individually optimized for each compound. For details refer to the Methods section.

**Method in-house validation.** The optimized LC-MS/MS method was thoroughly evaluated by analytical validation according to the guidelines established by the European Commission Decision Nº. 657/2002[29], with minor modifications. Urine, serum, and breast milk samples were chosen for validation and included extraction experiments at two trace-level concentrations. Overall, 75% and 79% of the 81 analytes fulfilled all criteria for at least one concentration level in urine and serum, respectively. For the more challenging breast milk matrix, the percentage was lower at 56%. For seven analytes, the selectivity, recovery or linearity were outside the guideline ranges in all matrices. Although these

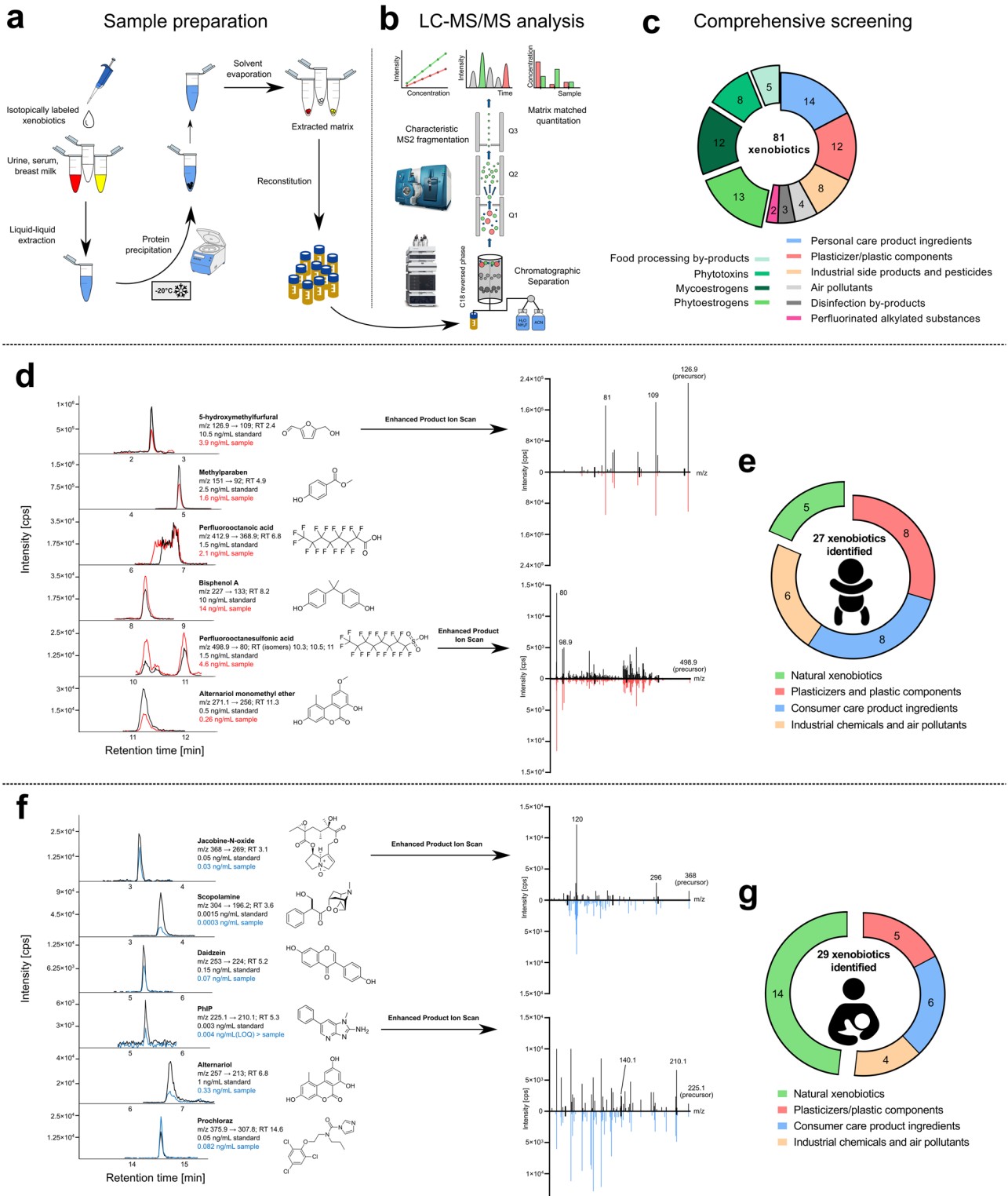

**Fig. 2 Analytical workflow, compound panel and identified analytes in neonatal plasma and breast milk. a**, **b** Overview of the generic sample preparation and LC-MS/MS workflow developed. **c** Included chemical classes. Chromatograms from scheduled multiple reaction monitoring (sMRM) show the transitions used to quantitate selected analytes in infant plasma and breast milk. Chromatograms for the corresponding matrix-matched standards are plotted in black, while the plasma (**d**) and breast milk (**f**) samples are displayed in red and blue, respectively. MS$^2$ spectra generated from enhanced product ion scans (EPIs) of reference standards are compared to the spectra acquired in biological samples for selected analytes (5-hydroxymethylfurfural, perfluorooctanesulfonic acid, jacobine-N-oxide and PhIP; technical details of EPI scans are provided in the Online Methods section). The precursor ion *m/z* is annotated together with fragment ions that were used for quantitative and qualitative measurements. Source data are provided as a Source Data file. Pie charts indicate the distribution of the detected compounds in neonate plasma (**e**) and breast milk (**g**) samples according to origin or chemical classification.

analytes are restricted to semi-quantitative or qualitative evaluation, the information obtained can still be valuable in environmental-health studies. Importantly, validation guidelines are still focused on tailored, single-analyte assays and typically do not encourage multi-analyte assays. As differing sensitivities between bio-fluids were expected in our next-generation HBM method, it was anticipated that not all compounds would completely fulfil the criteria at all chosen fortification levels in every matrix. Particularly, since fortification and calibration were based on the most sensitive matrix for each analyte (see Supplementary Table 5).

The extraction efficiencies from the urine and serum samples were high (median of 93% and 87%, respectively); while recoveries from breast milk were lower (median of 54%). Due to the high fat and protein content of milk, a two-phase extraction protocol was implemented to deplete the organic layer of matrix components. Thus, consistent with other reports[30], the expectation was that to a certain degree the polar analytes may be lost in the aqueous phase. Detailed results of the analytical validation are reported in Supplementary Table 3 and Supplementary Table 4. For most compounds, the limits of detection and quantitation (LODs/LOQs) were in the range of 0.001–0.1 ng mL$^{-1}$.

**Quantitating co-exposure in extremely premature infants**. In a proof-of-principle experiment, our analytical method was applied to heparin-derived plasma samples obtained 28 days post-partum from extremely premature infants (gestational age <28 weeks, birth weight <1 kg). Sample volumes were between 50 and 200 μL. For volumes <200 μL, the sample preparation procedure was adjusted accordingly. Additional spiking experiments verified the proper performance of the procedure even when done with 50 μL (see Supplementary Data 6). Detailed results are reported in Supplementary Data 1 and include hematocrit/albumin concentrations and results of three samples after enzymatic deconjugation for selected analytes (Supplementary Data 2). Figure 2d shows representative chromatograms of the transitions used to quantitate selected compounds. MS$^2$ fragment ion spectra are shown in addition to an authentic reference standard of similar concentration. As many as 27 xenobiotics were detected in at least one sample (Table 1; Fig. 2e); and included plasticizers/plastic components, industrial chemicals, phytoestrogens, mycoestrogens, personal-care product ingredients, food processing by-products and markers of exposure to smoke. Although our method was initially fully validated using serum and not plasma, the matrices were extensively and critically compared to demonstrate and ensure similar analytical behaviour. Further details, including a qualitative comparison of the plasma and serum matrices for selected analytes, are described in the Supplementary Information and Supplementary Fig. 1.

Interestingly, from the 27 identified chemicals, 22 originate from synthetic products and 12 thereof were detected in all the analyzed premature infant plasma samples. Of the detected analytes, those with the highest frequency included multiple plasticizers, industrial chemicals, parabens, PFOA and PFOS. Figure 3 shows a selection of the compounds that were identified and highlights the vast concentration range that is observed for some toxicants. A comparison of these concentrations with adult control samples revealed that the contamination of the infant samples with certain chemicals was substantially higher in many subjects. For example, 11 ng mL$^{-1}$ BPA, 12 ng mL$^{-1}$ PFOA, 17 ng mL$^{-1}$ 2-naphthol, 9.4 ng mL$^{-1}$ 4-tert-octylphenol, 0.13 ng mL$^{-1}$ ethylparaben, 15 ng mL$^{-1}$ methylparaben and 2.9 ng mL$^{-1}$ propylparaben were detected in one infant. These values are all higher than the concentrations determined for both adult controls. Clearly, the adults were not in contact with the same potential intensive care unit (ICU)-related exposure sources (see Discussion) as the neonate cohort. However, it is also known that infants have a reduced drug detoxification capability[31]. Consequently, with a lower proportion of conjugated species there might be a higher level of more potent xenobiotics in most of the participants in the cohort.

**Assessing early-life exposure via breast milk**. To demonstrate the feasibility of the method to determine chronic low-dose co-exposure in breast milk, the method was applied to 86 breast milk samples collected from a mother during the first 211 days post-partum. Individual daily results are reported in Supplementary Data 3 (without enzymatic hydrolysis) and Supplementary Data 4 (after enzymatic hydrolysis). A direct comparison of the two datasets is given in Supplementary Data 5. Chromatograms of the mass transition used for quantitation of selected compounds detected in the breast milk samples are plotted in Fig. 2f. Unexpectedly, 29 xenobiotics were detected in at least one sample (Table 1; Fig. 2g).

Figure 4 visualizes the exposure dynamics for toxicants that were detected in most samples. In particular, food-associated xenobiotics showed a highly dynamic behaviour. Detected levels spanned a wide concentration range over the observed period (i.e. <0.0083 ng mL$^{-1}$–0.14 ng mL$^{-1}$ for daidzein, Fig. 4d). In most cases, contamination with synthetic chemicals such as parabens (Fig. 4 a, b), phthalates (Fig. 4c) and PFAS (Fig. 4f) was less dynamic when compared to xenobiotics of natural origin. These observations are consistent with the known chemical- and biological properties (i.e. toxicokinetic half-lives, adduct binding in plasma), and routes of exposure. Phytoestrogen migration into breast milk varies in relation to nutritional intake. Paraben- and phthalate exposure may be associated with regular consumer care product usage and contact with plastics, apart from the few outliers indicating striking exposure on particular days. The bioaccumulative properties of PFAS are mirrored by the detectet levels of PFOS remaining relatively constant to the LOQ value on most days. Lastly, indicative of steady exposure, the background contamination of the PAH metabolite 2-naphthol also remained rather constant (Fig. 2e).

As a tool to reveal potential trends and associations between chemical exposures, a correlation matrix is shown in Supplementary Data 7. Most notably, a strong correlation between methylparaben and propylparaben levels is revealed, as well as the correlation between, among some others, ethylparaben and methylparaben/n-butylbenzenesulfonamide and glycitein and daidzein. This is in line with expected routes of exposure, as the respective compounds likely stem from identical or related sources.

To the best of our knowledge, the detection of the pyrrolizidine alkaloids jacobine- and riddelliin-N-oxide, the tropane alkaloids anisodamine and scopolamine, and the heterocyclic aromatic amine PhIP have never been previously reported in breast milk. The latter was detected at trace levels on two consecutive days of sampling and suggests consumption of a heavily contaminated food product, e.g. typically well-done, roasted or grilled meat. Phytotoxin contamination was apparent throughout the sampling period as the individual was known to regularly consume herbal tea mixtures that are intended to promote milk production. This study demonstrated that infants can be exposed to these plant alkaloids of emerging interest via breast milk if the mother regularly consumes certain foodstuffs or drinks.

**Estimation of daily toxicant intake of infants from breast milk measurements**. A high-exposure scenario was chosen to describe

**Table 1 Results of two independent proof-of-principle studies.**

| Compound | Infant plasma (n = 21) | | | Longitudinal breast milk (n = 86) | | |
|---|---|---|---|---|---|---|
| | Pos. samples | Min. [ng mL$^{-1}$] | Max. [ng mL$^{-1}$] | Pos. samples | Min. [ng mL$^{-1}$] | Max. [ng mL$^{-1}$] |
| **Plasticizers/plastic components** | | | | | | |
| Bisphenol A (BPA) | 21 | 0.72 | 15 | 2 | <0.40 | <0.40 |
| Bisphenol AF (BPAF) | 17 | <0.075 | <0.075 | — | | |
| Bisphenol F (BPF) | 2 | 0.091 | 0.39 | 1 | <0.028 | <0.028 |
| Bisphenol S (BPS) | — | | | 57 | <0.0078 | 0.051 |
| Mono-n-butyl phthalate (MBP) | — | | | 79 | <0.16 | 6.4 |
| Mono-2-ethylhexyl phthalate (MEHP)[a] | 21 | 0.062 | 75 | — | | |
| N-butylbenzenesulfonamide | 21 | 2.2 | 53 | 86 | 3.4 | 14 |
| Benzyl butyl phthalate | 21 | <0.66 | 2.7 | — | | |
| Dibutyl phthalate | 21 | <7.6 | 100 | — | | |
| Tetrabrombisphenol A (TBPA) | 1 | <0.33 | <0.33 | — | | |
| **Perfluorinated alkylated substances** | | | | | | |
| Perfluorooctanoic acid (PFOA) | 21 | 0.35 | 12 | 79 | <0.092 | <0.092 |
| Perfluorooctanesulfonic acid (PFOS) | 21 | <0.14 | 11 | 74 | <0.016 | 0.048 |
| **Industrial side products and pesticides** | | | | | | |
| 2-naphthol | 21 | 0.95 | 21 | 86 | 0.21 | 1.1 |
| Prochloraz | — | | | 1 | 0.082 | 0.082 |
| 4-tert-octylphenol (4-tert-OP) | 20 | <3.3 | 22 | — | | |
| Nonylphenol | 21 | 5 | 69 | — | | |
| **Phytoestrogens and metabolites** | | | | | | |
| 8-prenylnaringenin | — | | | 17 | <0.015 | 0.23 |
| Coumestrol | 11 | 0.0080 | 0.051 | — | | |
| Daidzein | — | | | 44 | <0.0083 | 0.14 |
| Enterodiol | — | | | 12 | 0.0079 | 0.025 |
| Enterolactone | — | | | 3 | <0.17 | <0.17 |
| Glycitein | — | | | 23 | <0.0023 | 0.011 |
| Isoxanthohumol | — | | | 14 | <0.0095 | 0.078 |
| Matairesinol | 4 | <0.11 | 0.26 | — | | |
| Resveratrol | — | | | 1 | <0.30 | <0.30 |
| Xanthohumol | — | | | 3 | <0.22 | <0.22 |
| **Mycoestrogens and metabolites** | | | | | | |
| Alternariol | — | | | 1 | 0.51 | 0.51 |
| Alternariol monomethyl ether | 11 | <0.012 | 0.26 | — | | |
| **Personal-care product ingredients, pharmaceuticals and metabolites** | | | | | | |
| Benzophenone 1 | 18 | <0.017 | 0.031 | 86 | <0.012 | 0.039 |
| Benzophenone 2 | — | | | 2 | <0.0096 | 0.020 |
| Benzylparaben | 2 | <0.0011 | <0.0011 | — | | |
| Butylparaben (BP) | 15 | <0.0010 | 0.33 | 14 | <0.0099 | 0.036 |
| Ethylparaben (EP) | 21 | 0.024 | 0.26 | 86 | <0.041 | 0.13 |
| Isobutylparaben | 7 | <0.0074 | 0.30 | — | | |
| Methylparaben (MP) | 21 | 0.061 | 15 | 86 | 0.069 | 23 |
| Propylparaben (PP) | 21 | 0.016 | 2.9 | 86 | 0.033 | 16 |
| Triclosan | 11 | <0.048 | <0.048 | — | | |
| **Phytotoxins** | | | | | | |
| Anisodamine | — | | | 9 | <0.0010 | 0.031 |
| Jacobine-N-oxide | — | | | 3 | <0.035 | 0.11 |
| Riddelliin-N-oxide | — | | | 1 | <0.021 | <0.021 |
| Scopolamine | — | | | 4 | <0.00014 | 0.0004 |
| **Food-processing by-products** | | | | | | |
| 5-Hydroxymethylfurfural (HMF) | 1 | 4.4 | 4.4 | — | | |
| 5-Hydroxymethyl-2-furanoic acid (HMFA) | 11 | <153 | 2555 | — | | |
| PhIP | — | | | 2 | <0.0042 | <0.0042 |
| **Air pollutants** | | | | | | |
| Cotinine[a] | 20 | 0.018 | 0.73 | — | | |

Chemical exposure identified and quantitated in a cohort of extremely premature infants (n = 21) and breast milk samples obtained from a breastfeeding mother during the first 211 days of life (n = 86). The number of identifications in each independent study are listed, as is the lowest (Min.) and highest (Max.) level of contamination. If the least contaminated sample was below the respective limit of quantitation (LOQ), it is indicated by '<'LOQ'.
[a]Quantitated using solvent calibration because the pooled matrix used for the matrix-matched calibration was heavily contaminated.

the intake of selected toxicants by infants through breast milk in the context of risk assessment. For calculations of a hypothetical upper bound intake (hUBI), average consumption of 1000 mL of breast milk per day and a body weight (bw) of 6 kg was assumed.

Toxicants that were detected below the LOQ were estimated at the LOQ level, while the highest detected concentration was assumed for quantitated exposures. Tolerable daily intake (TDI) or acceptable daily intake (ADI) values for adults were adapted

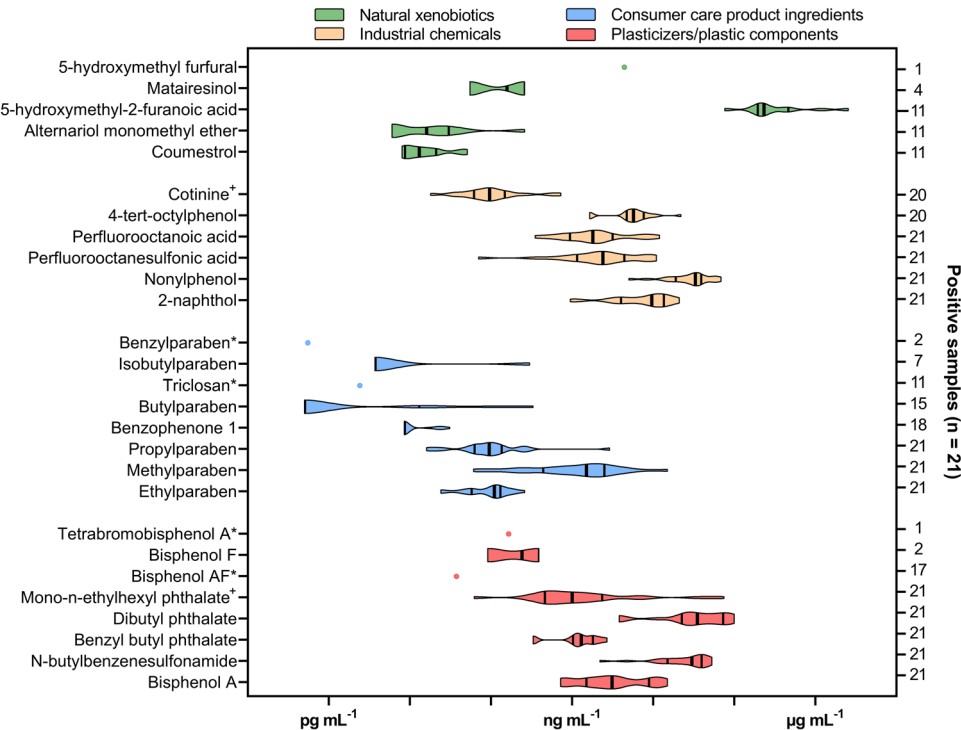

**Fig. 3 Distribution of chemicals detected in premature neonates.** Violin plots depict the distribution of detected concentrations of compounds in premature infant plasma samples ($n = 21$). Vertical bars inside the plot indicate the first quartile, the median and the third quartile. Analytes that were identified below the respective limit of quantitation (LOQ) were set to half the LOQ in this plot. Chemicals marked with (*) were only detected below the LOQ, thus no quantitative distribution is given and only a filled circle representing half the LOQ is displayed. Chemicals marked with (+) were quantitated using solvent calibration because the pooled serum intended for matrix-matched calibration was contaminated.

from the most current guidelines of the European Food Safety Authority (EFSA). For infants below 16 weeks of age, these recommendations were corrected by a factor of one-third of the TDI/ADI according to a general EFSA recommendation[32].

Mono-n-butyl phthalate (MBP) is the major metabolite of dibutyl phthalate with proven estrogenicity. A TDI of 10 µg kg⁻¹ body weight per day is defined for the parent compound (namely, 3.3 µg kg⁻¹ body weight per day for infants), but not yet determined for the metabolite[33]. Even if a similar TDI were assumed for MBP, our hUBI of 1.1 µg kg⁻¹ body weight per day would not exceed this recommendation. Similarly, TDI of BPA (1.3 µg kg⁻¹ body weight per day) was not exceeded in our model. Even the cumulative hUBI of all detected bisphenols would only result in a cumulative hUBI of 0.08 µg kg⁻¹ body weight per day for bisphenols. However, EFSA proposed in 2021 to significantly reduce the TDI for BPA which would result in exceedances.

Except for the widespread and potent representative scopolamine, research on the risk assessment of tropane alkaloids is scarce. Here, the TDI is not exceeded. For pyrrolizidine alkaloids the EFSA does not define any TDIs or ADIs currently. Due to a lack of data, the risk assessment for these food contaminants is based on the calculation of a cumulative 70 µg kg⁻¹ body weight per day benchmark dose lower confidence limit (BMDL₁₀) for a 10% excess cancer risk. From a public-health perspective, the EFSA panel concluded that a margin of exposure (MoE) above 10,000 is of low concern for this group of phytotoxins[34]. For jacobine-N-oxide, our data suggest that contaminated breast milk may substantially contribute to this pyrrolizidine alkaloid burden resulting in a MoE of 5000. More conservative guidelines, such as a TDI defined by the Dutch National Institute of Public Health and the Environment of 0.1 µg kg⁻¹ body weight per day would indicate a MoE of 1.8 after age correction for infants (0.033 µg kg⁻¹ body weight per day)[34].

These findings demonstrate that there is a potential health risk for infants of mothers that consume high quantities of pyrrolizidine alkaloids via honey or herbal remedies.

The EFSA has not established any TDIs for heterocyclic aromatic amines (HCAs) such as PhIP because no health thresholds can be derived for a carcinogen. Generally, higher intake is associated with increased cancer risk. Current epidemiological data on the dietary exposure to HCAs are inconsistent, varies between populations, and data specifically on PhIP intake is rare. Low estimates determined from a Croatian female population[35] indicate a PhIP intake above 2 ng kg⁻¹ body weight per day. Our estimated hUBI at 7 ng kg⁻¹ body weight per day demonstrates the possibility of higher early-life exposure of infants via breast milk. A proposed BMDL₁₀ of 480 µg kg⁻¹ body weight per day for the development of prostate tumours, however, remains unattainable.

In accordance with available guidance values (Table 2), for most xenobiotics, breast milk does not act as a vehicle for contamination levels of concern for the infant. This is consistent with another recent report highlighting mycotoxin levels in breast milk substitutes and comparing it to potential exposure through breast milk[36]. According to our estimates, only pyrrolizidine alkaloid levels may warrant further risk assessment. For many classes, e.g. phytoestrogens, *Alternaria* toxins and BPA substitutes; however, no official health-based guidance values have been published to date and phytoestrogens are typically associated with beneficial effects during early life.

## Discussion

A comprehensive and generic LC-MS/MS assay to simultaneously identify and quantitate multiple classes of endocrine-disrupting and other potentially harmful toxicants and contaminants (81 xenobiotics in total) was developed and the performance of the

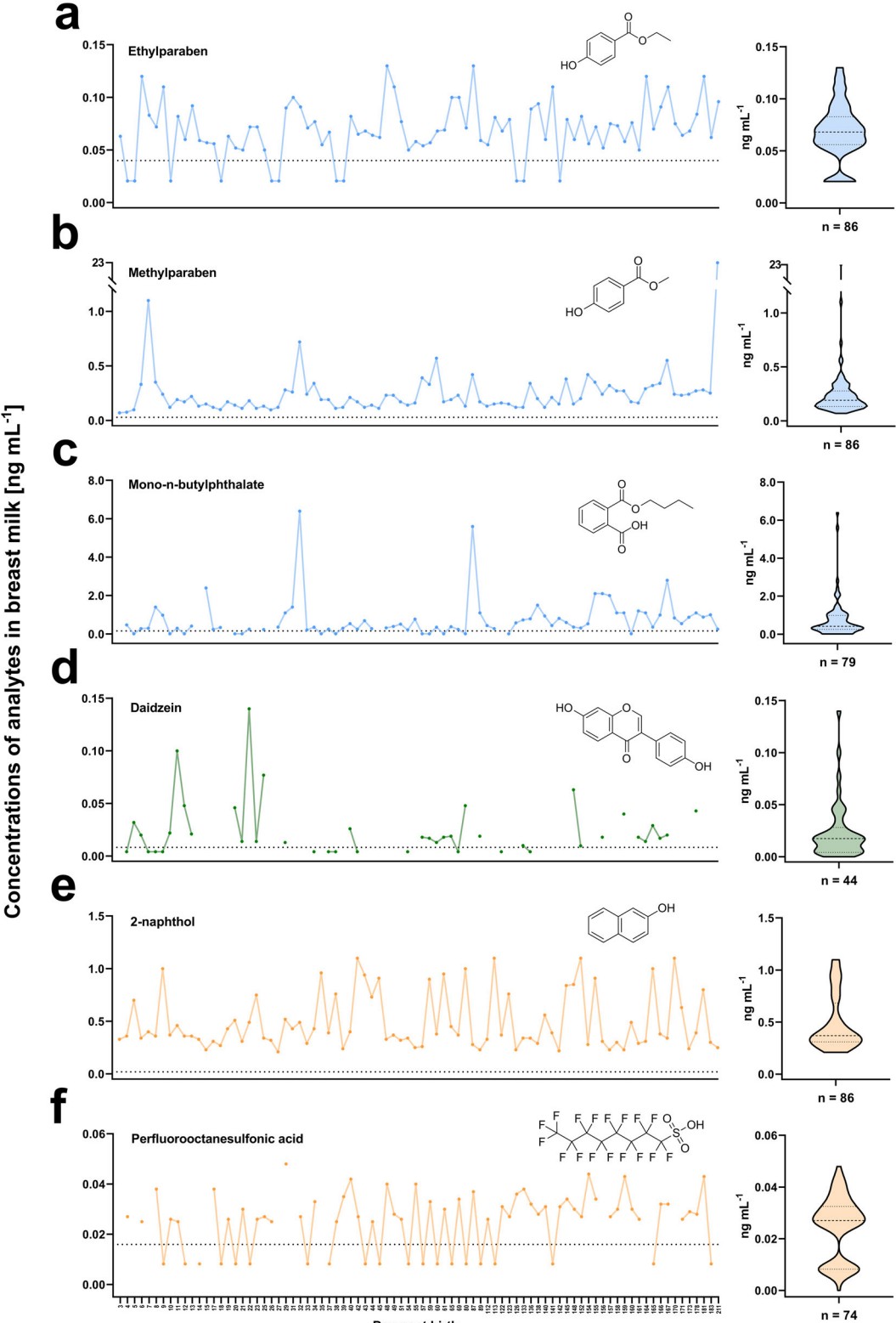

**Fig. 4 Exposure dynamics of selected xenobiotics identified in breast milk. a–f** Detected levels of ethylparaben, methylparaben, mono-n-butyl phthalate (plastic- and consumer product-associated, depicted in blue), daidzein (food-associated, depicted in green), 2-naphthol and perfluorooctanesulfonic acid (industrial- and/or environment-associated, depicted in orange), respectively. Samples were taken between day 3 and 211 post-partum ($n = 86$). The horizontal dotted line indicates the limit of quantitation (LOQ) for the respective compound. Samples were acquired over an irregular schedule (marked on the x-axis) as the infant was fed before sampling and, in some cases, no breast milk remained for laboratory analysis. Dots connected by lines indicate detection on consecutive days of sampling. Analytes that were determined below the respective LOQ values were set to half the LOQ for calculations and plotting. Violin plots depict the distribution of concentrations over the observed period. Horizontal bars inside the plot display the first quartile, the median and the third quartile. The number of positive samples is reported on the x-axis for each respective analyte.

**Table 2 Preliminary exposure/risk assessment using a high-exposure scenario.**

| Compound | Assumed concentration in breast milk [ng mL$^{-1}$] | TDI or ADI adults [µg kg$^{-1}$ bw day$^{-1}$] | TDI or ADI infant corrected [µg kg$^{-1}$ bw day$^{-1}$] | hUBI [µg kg$^{-1}$ bw day$^{-1}$] | Ratio TDI or ADI/ hUBI or MoE |
|---|---|---|---|---|---|
| **Plasticizers/plastic components** | | | | | |
| Bisphenol A (BPA) | 0.40 | 4.0[52] | 1.3 | 0.067 | 20 |
| Bisphenol F (BPF) | 0.028 | | | 0.0047 | |
| Bisphenol S (BPS) | 0.051 | | | 0.0085 | |
| Mono-n-butyl phthalate (MBP) | 6.4 | | | 1.1 | |
| **Perfluorinated alkylated substances** | | | | | |
| Perfluorooctanoic acid (PFOA) | 0.092 | 0.63[53] | 0.21 | 0.015 | 14 |
| Perfluorooctanesulfonic acid (PFOS) | 0.048 | 0.63[53] | 0.21 | 0.0080 | 26 |
| **Industrial side products and pesticides** | | | | | |
| Prochloraz | 0.082 | 100[54] | 33 | 0.014 | 2439 |
| **Phytoestrogens and metabolites** | | | | | |
| 8-prenylnaringenin | 0.23 | | | 0.038 | |
| Daidzein | 0.14 | | | 0.023 | |
| Enterodiol | 0.025 | | | 0.042 | |
| Enterolactone | 0.17 | | | 0.028 | |
| Glycitein | 0.011 | | | 0.0018 | |
| Isoxanthohumol | 0.078 | | | 0.013 | |
| Resveratrol | 0.30 | | | 0.050 | |
| Xanthohumol | 0.22 | | | 0.037 | |
| **Mycoestrogens and metabolites** | | | | | |
| Alternariol | 0.51 | | | 0.085 | |
| **Personal-care product ingredients, pharmaceuticals and metabolites** | | | | | |
| Benzophenone 1 | 0.039 | 30[55] | 10 | 0.0065 | 1538 |
| Benzophenone 2 | 0.020 | | | 0.0033 | |
| Butylparaben (BP) | 0.036 | 2000[56] | 667 | 0.0060 | 111,111 |
| Ethylparaben (EP) | 0.13 | 10,000[56] | 3333 | 0.022 | 153,846 |
| Methylparaben (MP) | 23 | 10,000[56] | 3333 | 3.8 | 870 |
| Propylparaben (PP) | 16 | 2000[56] | 667 | 2.7 | 250 |
| **Phytotoxins** | | | | | |
| Anisodamine | 0.031 | | | 0.0052 | |
| Jacobine-N-oxide | 0.11 | BMDL$_{10}$: 70[34] | | 0.018 | 3889 |
| Riddelliin-N-oxide | 0.021 | BMDL$_{10}$: 70[34] | | 0.0035 | 20,000 |
| Scopolamine | 0.0004 | 0.016[57] | 0.0053 | 0.000067 | 79 |
| **Food-processing by-products** | | | | | |
| PhIP | 0.042 | BMDL$_{10}$: 480[58] | | 0.0070 | 68,571 |

The calculated hypothetical upper bound intake (hUBI) is compared to currently recommended tolerable daily intakes (TDI) or acceptable daily intakes (ADI) for adults and corrected for infants below 16 weeks of age. The margin of exposure (MoE) describes the ratio of a calculated benchmark dose lower confidence limit for a 10% excess cancer risk (BMDL10) to the hUBI for pyrrolizidine alkaloids and PhIP.

assay critically evaluated. Alongside the assessment of the method's linearity, sensitivity and selectivity, two trace concentration levels were applied in spiking experiments to determine analyte recoveries. The method yielded highly convincing results during validation for urine and serum, resulting in more than 75% of the chemicals included fulfilling all criteria for at least one of the fortification levels. This was despite the use of low sample volumes and the application of stringent guidelines intended for targeted single-analyte assays. Even for breast milk, a complex, high-fat matrix, the results were acceptable (56%). Importantly, compounds that did not pass all criteria may still be assessed in a semi-quantitative manner to potentially draw initial conclusions in future ExWAS.

The main shortcoming of most current targeted HBM methods is the limited number of toxicants that are included. These are usually restricted to fewer than 20 analytes that belong to a single-chemical group of either naturally occurring or synthetic chemicals. It is challenging to design appropriate sample preparation schemes and chromatographic- and mass spectrometric parameters to enable the concurrent analysis of a multitude of structurally different compounds at trace levels. Consequently, multiple single-class methods need to be employed to achieve a

broad assessment. This comes with high cost, additional labour, longer analysis times and the need for a higher sample volume.

Premature infants are at very high risk of suffering from life-long neurological morbidities, including neurosensory impairments, cerebral palsy, cognitive and motor delay and attention deficit hyperactivity disorder[37]. It is widely accepted, that aberrant inflammation due to post-natal infection can drive the pathogenesis of brain damage[38], but the xenobiotic influence on early-life immunological development remains poorly explored. Our previously established assay that focused on endogenous and exogenous estrogens was able to assess a comparatively high number of xenobiotics[23]. However, the selection was based on estrogenic potential and did not include important key toxicants targeting other modes of action including genotoxicity, carcinogenicity, immunotoxicity or neurotoxicity. The expanded panel of analytes includes plasticizers, perfluorinated alkylated substances, industrial side products, phyto- and mycotoxins, personal-care product ingredients, disinfection by-products, food-processing by-products and markers of exposure to (cigarette) smoke. We found for instance, that the neonates in our cohort were widely exposed to 2-naphthol, a compound shown to negatively affect the expression of thyroid hormones, thereby potentially

impairing early-life brain development[39]. Also, we found perfluorinated compounds to be widely distributed in both premature infant plasma, as well as in breast milk. PFAS were shown to cause persistent neurological defects in adult mice after exposure during the neonatal phase[40]. The possibility of such exposome-scale analysis will enable the assessment of complex co-exposure effects to unravel potential clinical associations between these highly different chemical classes in the future. Besides the broader coverage of analytes and toxicant classes, the presented assay is additionally far more sensitive (on average a factor of ~60 × for the analytes included in ref. [23] method); the currently most pressing need for generating meaningful data in exposomics. This capacity was clearly demonstrated e.g. by the identification and quantitation of several highly potent xenobiotics of general interest in human breast milk that has not been described before in this matrix (PhIP, pyrrolizidine- and tropane alkaloids). Importantly, the instrumental platform demonstrated striking linearity, accuracy and precision at the described trace levels that have not been reached before at this scale.

The high sensitivity of our method with LOD values mostly <0.05 ng mL$^{-1}$ generally outperformed previously published, more tailored multi-class assays, that feature a far smaller number of xenobiotics (see Supplementary Table 6). This was possible by the application of an effective chromatographic separation, NH$_4$F as an ionization additive and the latest generation LC-MS instrumentation. Our approach employs a simple and rapid, yet comprehensive, sample extraction procedure that enabled the concurrent detection of almost 30 highly diverse exposures in blood and breast milk samples obtained from real-life studies. In particular, breast milk posed an analytical challenge due to the complexity of the matrix. Thereby, we demonstrated that if the newest generation of LC-MS/MS instrumentation is utilized in a more holistic manner, non-tailormade methods to analyze chronic exposure levels are feasible. One drawback of targeted analyses is, however, that conjugated chemicals are often not included in mass transition lists and are therefore overlooked. Hence, the developed LC-MS/MS method also includes some glucuronides and sulfates; albeit only a limited number of analytes are included as reference standards of phase-II metabolites are rarely commercially available. As the volumes of the available materials were mostly sufficient for only one sample preparation, an enzymatic deconjugation of the neonate plasma samples was only tested for a few samples in this work. In addition, all breast milk samples were enzymatically treated and re-measured. However, after the application of a crude enzyme extract (β-glucuronidase and sulfatase from *Helix pomatia*) increased baseline noise was observed. As the method was analytically validated without enzymatic treatment, this resulted in issues with selectivity and sensitivity (i.e. previous detections could not be replicated) for some compounds. Further, the enzyme solution was shown to be contaminated with many of the included analytes, resulting in the introduction of contamination. Nonetheless, the experiments confirmed the assumption that, for chemicals where the newly introduced enzyme-matrix did not result in severe analytical interference, increased toxicant concentrations can be obtained. In future studies, this approach has the potential to increase the number and concentrations of the xenobiotics detected, most notably in urine, following thorough re-validation. The use of proteolytic enzymes (e.g. pronase) in future work might be considered if highly electrophilic molecules with an affinity for albumin/protein adducts are in focus.

The readily accessible bio-fluids urine and blood are the most commonly used matrices in established and automated sample preparation strategies[41]. Independent of the route of exposure, ingested compounds are typically anticipated in blood because this bio-fluid is in steady-state contact with most human tissues.

Therefore, it may be considered the best-suited matrix to characterize the chemical exposome[42]. In parallel, complementary urinary analyses can shed light on short-lived compounds or excreted metabolites. Colorimetric assays to determine creatinine in urine and albumin in the blood are typically implemented to report xenobiotic levels in relation to the respective bio-fluid concentrations (see Supplementary Data 1).

However, it is vital to critically question which toxicity-related questions can be answered by the data created using such exposome-scale HBM methods. This clearly depends on the investigated biological matrix and the chemical properties of a specific toxicant. Blood may be a suitable matrix to assess chronic contamination with persistent organic pollutants (e.g. PFAS), while for immediately metabolized and excreted compounds such as phthalates, bisphenols or parabens, the parent compounds are more of a proxy of acute—rather than chronic ingestion[43]. For such chemicals that do not feature any persistent metabolites, longitudinal urinary analysis of the predominant phase-II metabolites over longer periods of time, either by a direct assessment or by the application of deconjugation, may be better suited to assess chronic and dynamic exposures. Moreover, the method does not yet include direct markers of biological effect. Consequently, we believe this work is of prime relevance in the area of exposure monitoring, rather than biological effect monitoring. This distinction needs to be stated clearly, as the former may be used to only find associations between disease development and exposure patterns. We encourage the generation of sufficiently large datasets by exposome-scale investigations to increase statistical power. Biomarkers of biological effect directly assess the adverse effect of a xenobiotic. For many chemicals no specific biomarkers of effect have been defined as their multidimensional mechanisms (e.g. EDCs acting on multiple fronts in the hormonal system) are not fully elucidated. For adduct-forming toxicants (e.g. acrylamide, PAHs, PhIP or phytotoxins) the availability of analytical standards severely limits the inclusion of DNA- or protein conjugates to date.

These issues highlight an important advantage of the presented approach, namely its extendable design. At the technical level, the LC-MS/MS system is still far from working at full capacity, as it still holds the potential of the inclusion of hundreds of additional mass spectrometric transitions. With future advances concerning the knowledge of emerging toxins or relevant biotransformation products as well as the availability of appropriate analytical standards, each compound category might be further expanded with additional markers of exposure or direct biomarkers of biological effect. While measuring the entire exposome by a single analytical method or platform will not be feasible, we see the presented approach as an ambitious attempt to cover a vast chemical space that can be further expanded.

The cohort of premature infants evaluated in this study is representative of a susceptible sub-population and the data obtained revealed the presence of several xenobiotics, some at high levels. PFAS are known to readily pass from maternal blood to the developing embryo and are associated with multiple adverse health outcomes during pregnancy, including low birth weight caused by impaired placental function[44]. Potential sources of exposure to plasticizers include contact with medical equipment such as ventilators, feeding tubes and infusion pumps in the neonatal ICU. High paraben levels may derive from the injection of pharmaceuticals that include them as excipients or preservatives[45]. The detection of cotinine in 20 individuals from the cohort points towards the possibility of systemic passive exposure to smoke. At first glance, this was a particularly unexpected outcome. Although it was not possible to accurately quantitate cotinine using matrix-matched calibration (high contamination of the commercially available pooled human serum

used for calibration purposes); by applying solvent calibration, levels <0.5 ng mL$^{-1}$ were estimated for all samples. This is consistent with previous findings of mean cotinine levels in non-smokers as a result of everyday exposure to passive smoking[46]. The contamination of donated, pooled breast milk that is routinely used as a supplement in the clinic may be a source of this ubiquitous contamination, although active smokers are excluded as milk donors. This hypothesis is reinforced by the identification of both nicotine metabolites, cotinine and trans-3-hydroxy cotinine, in the breast milk that was applied for matrix-matched calibration (Supplementary Fig. 1). However, this breast milk pool was provided by a different clinic and was not the pool that had been fed to the neonate cohort (see Methods). Although synthetic xenobiotics comprise the majority of the quantitated chemicals, the simultaneous detection of HMF and HMFA in one individual demonstrated the successful quantitation of a food carcinogen and the main human metabolite thereof. We surmised that the pasteurization process of the donor milk during preparation for the premature babies may have led to the pyrolytic formation of HMF; and the infant was subsequently exposed during ingestion. It is highly likely that contaminated milk may have also resulted in the ingestion of a limited number of phyto- and mycoestrogens at low concentrations.

There are only a few defined critical threshold values concerning blood levels of chemical contamination. Current human biomonitoring values defined by the German HBM-Commission are based on epidemiological studies and animal experiments and are defined as the concentration below which no adverse health effects are expected. For PFOA and PFOS, these values are 2 and 5 ng mL$^{-1}$, respectively[47]. The reference values were exceeded in the plasma samples from multiple infants. Although no maximum recommended dosages of parabens in blood have yet been proposed using epidemiological data; in a recent ExWAS[10], Agier et al. concluded that paraben exposure is a health risk during early life. Of five identified parabens, propyl-, methyl- and ethylparaben were contaminations in all premature babies. While previous assessments focused almost exclusively on urinary analyses of phthalates[20], ubiquitous contamination of blood was also observed in our data. As mentioned above, this matrix may be more amenable to elucidating the acute biologically-active burden of such short-lived compounds that causes a negative outcome. Consequently, quantitative data from blood analyses may also be more suitable for the discovery of associated biological effects (i.e. inflammation, oxidative stress and endocrine disruption) and the subsequent proposal of reference values.

Breast milk has been previously utilized to monitor early-life exposure of infants. This bio-fluid may act as a vehicle to transport unintended contaminants from mothers to offspring[31]; who are particularly susceptible to adverse biological effects because of a less-developed immune system[32] and less effective drug detoxification capabilities[31]. Our data clearly confirmed earlier findings that breast milk is a source of exposure for a child to endocrine disruptors and carcinogenic food contents during early life. Excluding the pyrrolizidine alkaloids, however, if single-chemical exposure is only considered, our estimated upper boundary for daily exposure levels should not pose a health risk for infants. Importantly, the longitudinal proof-of-principle study only included a single mother since this highly time-resolved sampling design is difficult to implement more broadly. While we focused on the potential contribution of breast milk toward the total xenobiotic burden of a neonate in this work, it must be clearly stated that low-level chemical exposure through breast milk must never be a factor to reduce or avoid breastfeeding. We recently demonstrated that alternatively consumed infant formula likely leads to higher mycotoxin exposure than the consumption of breast milk[36]. This might be also true for other toxicant classes.

Most importantly, the benefit of antibodies, growth factors and bioactive compounds including oligosaccharides in breast milk must be explicitly highlighted[48].

The full potential of adverse health effects caused by the co-occurrence of numerous groups of chemicals of interest in this particularly vulnerable cohort cannot yet be fully judged. As highlighted above, the true cumulative implications of the detected contaminations with respect to the development of chronic disease, particularly when considering the endocrine-disrupting or genotoxic potential of several co-occurring xenobiotics, still remain elusive.

In conclusion, this study not only demonstrates the pressing need for systematic, large-scale assessment of human xenobiotic exposure but also the technical feasibility of innovative analytical workflows. Importantly, the developed platform is generic in nature and thus enables continuous expansion with additional chemicals for exposure- or direct effect monitoring. Further, it is not restricted by small volumes as exemplified by the highly precious samples from the extremely premature infant cohort. Xenobiotic classes are not limited to food and environmental toxicants, but may also include biotransformation products thereof, drugs of abuse, microbiome-related compounds, or drugs of interest for the elucidation of drug-exposome interactions[49]. Improved characterization of the chemical exposome will not only contribute to the prevention of pathogenesis at a public and personalized level; but it is also intended to aid in the therapy of developing and existing conditions. We encourage the introduction of such comprehensive and highly-sensitive biomonitoring methods, routine monitoring schemes during all stages of life and true exposome-scale epidemiological studies investigating conditions such as neurodegenerative diseases, metabolic disease or different types of cancer.

## Methods

**Sample collection.** The performed research complied with all relevant ethical regulations and was approved by the ethics committee of the University of Vienna (no. 00157) and the ethics committee of the Medical University of Vienna (no. 1348/2017). For the method validation and matrix-matched standards, commercially available pooled AB plasma-derived serum (Sigma–Aldrich) was utilized following a comparison to heparinized pooled AB plasma (Innovative Research). The pooled urine was provided by a female volunteer who reduced exposure to xenoestrogen by avoiding foods or cosmetics stored in plastic containers and foods rich in phytoestrogens for 2 days prior to sample collection. Pooled breast milk was provided by the Semmelweis Women's Clinic in Vienna[50]. Urine and breast milk matrices were stored at −20 °C and the blood-derived matrices were stored at −80 °C.

Blood plasma from extremely prematurely born infants (gestational age <28 weeks and birth weight <1 kg) collected on day 28 after delivery were provided by the Department of Neonatology, Pediatric Intensive Care and Neuropediatrics at the Medical University of Vienna and stored at −80 °C as part of a larger microbiome-focused study[51]. Of the 11 male and 10 female subjects, 13 were delivered by caesarean section and 8 vaginally. Gestational ages ranged 25.6 ± 1.2 weeks and discharge ages ranged 39.0 ± 3.4 weeks. Birth weights ranged 780 ± 149 g and discharge weights ranged 2860 ± 657 g. All parents gave written-informed consent prior to study inclusion. Spanning a period of 3–211 days post-partum, 86 pooled breast milk samples were collected from one 31-year-old mother and stored at −20 °C until analysis. Multiple samples of pumped breast milk were collected throughout a day and partially up to 48 h with interim storage at 4 °C, and combined into an aggregate sample. This was done depending on the needs of the mother and the infant, i.e. left-overs not consumed by the infant were mixed at different times of the day. This approach was deemed the best option to obtain representative, pooled samples without interfering with the breast milk consumption of the infant as pointed out before by ref. [50] who examined these samples specifically on mycotoxin exposure. On some days, all breast milk was needed for feeding the infant, leaving no sample for laboratory analysis. Informed consent was obtained from all human research participants.

**LC-MS/MS method development and analysis.** Experiments were performed using a 1290 Infinity II LC system (Agilent) equipped with a VanGuard (1.8 μm) pre-column and an Acquity HSS T3 (1.8 μm, 2.1 × 100 mm) reversed-phase column (both Waters) coupled to a QTrap 6500+ mass spectrometer equipped with a Turbo-V™ ESI source (both Sciex) using fast polarity switching. The LC method

was adapted from Preindl et al. (2019)[23] and used water with 0.3 mM ammonium fluoride as aqueous eluent A and acetonitrile (ACN) as organic eluent B. Sample volumes of 5 μL were injected and a flow rate of 0.4 mL min$^{-1}$ was employed. Elution began with 5% B from 0–1 min; increased to 18% B until 1.8 min; then further increased to 35% B until 4.2 min; followed by an increase to 48% B until 13 min and a final increase to 90% B with constant elution until 15.8 min. The column was then flushed with 98% B until 17.6 min and re-equilibrated with 5% B from 17.7 to 20 min. The column compartment and the autosampler were maintained at 40 and 7 °C, respectively. The compound-dependent parameters for the multiple reaction monitoring (MRM) experiments and further source parameters were optimized. Detailed information is provided in the Supplementary Information and Supplementary Table 2.

Enhanced product ion scans were conducted to confirm the presence of selected analytes. MS$^2$ spectra were recorded from $m/z$ 50 to the precursor ion mass of the respective analyte + 2 $m/z$ with a scan rate of 1000 $m/z$ per second. The following CEs were utilized for precursor fragmentation: 20V, −70V, 45V and 50V for HMF, PFOS, jacobine-N-oxide and PhIP, respectively. No CE spread was used. The dynamic fill time mode and an excitation energy AF2 of 0.1V were utilized. The remaining source parameters were similar to those utilized in the validated LC-MS/MS method (Supplementary Table 2). Scans were conducted in profile mode and spectra were then converted to centroid mode.

**Sample preparation**. All sample preparation was conducted in plastic Eppendorf tubes on ice to minimize potential compound degradation. For the urine and serum/plasma samples, 200 μL were fortified with 20 μL of an internal standard mixture containing 13 isotopically labeled xenobiotics in ACN/H$_2$O (10/90, v/v) (Supplementary Table 2). Twenty microlitres ACN/H$_2$O (10/90, v/v) were added (20 μL of a solvent standard in ACN/H$_2$O (10/90, v/v) were used in spiking experiments for method validation), followed by 770 μL ACN/MeOH (1/1, v/v). For extraction, samples were sonicated for 10 min on ice. Proteins were precipitated for 2 h at −20 °C and the samples were then centrifuged for 10 min at 18,000 × $g$ at 4 °C. The supernatant was transferred and dried in a vacuum concentrator at 4 °C overnight. The residue was reconstituted in 200 μL ACN/H$_2$O (10/90, v/v). Afterwards the samples were again vortexed and centrifuged for 10 min at 18,000 × $g$ at 4 °C. The supernatant was finally transferred to amber glass vials for LC-MS/MS measurement and stored at −20 °C until analysis.

Due to the high fat and protein content, a different sample preparation protocol was used for the breast milk samples (accredited to ref. [30]). 250 μL aliquots were fortified with 25 μL of the internal standard mixture in ACN/H$_2$O (10/90, v/v), followed by the addition of 25 μL ACN/H$_2$O (10/90, v/v; 25 μL of a solvent standard in ACN/H2O (10/90, v/v) were used in spiking experiments for method validation). To denature proteins, 250 μL 1% formic acid in ACN were added and samples were vortexed for 3 min. The solution was added to 100 mg anhydrous MgSO$_4$ and 25 mg NaCl and again vortexed for 3 min, followed by centrifugation at 18,000 × $g$ for 10 min at 4 °C. The upper organic phase was stored at −20 °C for at least 2 h to precipitate proteins and again centrifuged at 18,000 × $g$ for 10 min at 4 °C. The supernatant was then lyophilized with the vacuum concentrator at temperatures between 4 and 15 °C until dry. The residue was reconstituted in 250 μL ACN/H$_2$O (10/90, v/v), vortexed and centrifuged at 18,000 × $g$ for 10 min at 4 °C. The supernatants were transferred to amber glass vials for LC-MS/MS measurements and stored at −20 °C until analysis.

To verify that downscaling the sample volume from 200 μL to 50 μL is feasible if only a limited amount of sample is available (as was done with the neonate cohort), we processed three samples of spiked plasma and compared the quantitative results for selected, representative analytes. As reported in Supplementary Data 6, the results remain similar with small standard deviations, confirming that values do not differ significantly in the two volumes tested.

For additional deconjugation experiments in breast milk samples, 100 μL of buffered β-glucuronidase/arylsulfatase (from *Helix pomatia*; activity = 2000 U/mL (glucuronidase) and 21 U/mL (sulfatase)) were added to 100 μL of breast milk and incubated at 37 °C for 17 h. For the neonate cohort, selected plasma samples were adjusted to the remaining plasma (40 μL sample + 40 μL enzyme solution) as sample volumes were limited and only available for three neonate samples.

**Method validation**. The method was in-house validated for all three matrices according to the European Commission Decision No. 657/2002[29]. Three independent sample preparations were performed with two weeks between each measurement series. Matrix-matched calibration standards were prepared by dissolving the extracted blank sample matrix in the respective solvent standards after evaporation. The multi-analyte standard for calibration was based on the matrix with the highest sensitivity for each analyte. As no certified reference material was available, spiking experiments at 3× the approximate LOQ (limit of quantification defined as a signal-to-noise ratio of 10) and 30× the LOQ were conducted in triplicates. LOQs and LODs (limit of detection defined as a signal-to-noise ratio of 3) and matrix effects in the form of signal suppression or enhancement (SSE evaluated by the ratio of the slope of the matrix-matched calibration and the slope of the solvent calibration) were evaluated once per validation run ($n = 3$). Linearity was assessed by the regression coefficient R$^2$ ($n = 3$), but as no specific value is stated in the guidelines, a limit of R$^2$ > 0.9 was assumed. Intermediate precisions

were calculated as the relative standard deviation between the results of the spiking experiments ($n = 9$). To assess selectivity, matrix blanks and system blanks were prepared in triplicates by extracting sample matrix and LC-MS grade water, respectively, and screened for interfering peaks. Method repeatability (intra-day precision) was evaluated by the repeated measurement of one validation batch ($n = 6$).

The validation guideline suggests applying the Horwitz Equation to calculate the minimum intermediate precision. For most of the spiking concentrations used in this method (<0.1 ng mL$^{-1}$), employing this equation yields tolerated deviations that were comparably high, i.e. 45% (1 ng mL$^{-1}$), 64% (0.1 ng mL$^{-1}$) and 91% (0.01 ng mL$^{-1}$). Therefore, the decision was made to apply stricter limits. The highest acceptable deviation for the intermediate precision was thus defined as 32% for concentrations <10 ng mL$^{-1}$, 27% for concentrations between 10 and 100 ng mL$^{-1}$, 23% for concentrations between 100 and 1000 ng mL$^{-1}$ and 16% for concentrations >1000 ng mL$^{-1}$. The same limits were used for the repeatability. A recovery range (defined as the proportion of the detected concentration to the spiked amount that had been added to the blank matrix prior to sample preparation) of 50–120% was accepted for all analytes independent of the spiked concentrations because the quantitative results of the unknown samples were corrected by the respective average recovery.

**Quantitation, data evaluation and figures**. Data evaluation was performed using quantitation software included in Sciex OS (version 1.6, Sciex). A weighted linear-regression model (1/×) was selected. Additional calculations were conducted in Excel (version 2010, Microsoft). Refer to the Supplementary Information for analytes that were internally calibrated. Unknown samples were quantitated using the corresponding matrix-matched calibration and results were only corrected with the respective average extraction recovery (determined during method validation) if the detected concentrations were above the average LOQ calculated after method validation, or featured a signal-to-noise ratio >10.

Importantly, results of unknown samples, as well as results of method validation, were additionally corrected by the system contamination with certain compounds that most likely occurred as extractables and/or leachables from the used laboratory tubes during sample preparation and/or the LC instrumentation itself (BPA, BPS, MBP, MEHP, benzyl butyl-/dibutyl phthalate, n-butylbenzenesulfonamide, nonylphenol). This was done by applying triplicate analytical blanks (extracting water) and determining the extent of the contamination via solvent regression.

Chemical structures were drawn using ChemDraw (version 12.0.2; PerkinElmer), plots were generated using GraphPad PRISM (version 8.0.1; Graphpad Software, Inc.) and figures were constructed in Inkscape (version 1.0.1; Inkscape).

**Reporting summary**. Further information on research design is available in the Nature Research Reporting Summary linked to this article.

## Data availability
The authors declare that the data supporting the findings of this study are available within the article and its Supplementary Information files. Raw data of the LC-MS measurements of the two study cohorts are accessible via the MetaboLights repository (https://www.ebi.ac.uk/metabolights), under identifier MTBLS2442. Additional data are available from the corresponding author upon request.

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

## Acknowledgements

We thank all volunteers for their participation, all medical staff for their help with sample collection and the members of the Warth lab for critical discussions and feedback. The Mass Spectrometry Centre (MSC) of the Faculty of Chemistry at the University of Vienna and Sciex (Darmstadt, Germany) are acknowledged for providing mass spectrometric instrumentation. This work was performed with the financial support of the University of Vienna, an inter-university cluster project grant between the University of Vienna and the Medical University of Vienna ("PreMiBrain"), the Austrian Science Fund (FWF, P33188) and the European Research Council (Starting Grant: FunKeyGut 741623).

## Author contributions

T.J. and B.W. conceived and designed the experiments. T.J. performed the mass spectrometric experiments. T.J., M.F. and D.Br. performed data analysis and interpretation. D.S., D.Be., A.B. and L.W. provided plasma samples and critical input. Y.F. and D.W. conducted

additional spiking and digestion experiments and related data evaluation. B.W. supervised the study and supported analyses and data evaluation/interpretation. All authors contributed to manuscript writing and have given approval to the final version of the paper.

## Competing interests

The authors declare no competing interests.
