## [Peer Review File · Nature Communications]

Reviewer comments, first round -

Reviewer #1 (Remarks to the Author):

This work aims at the measurement of a panel of diverse xenobiotics in blood, urine and breast milk by liquid chromatography coupled to tandem mass spectrometry. It should be seen as a proof of principle study showing that highly diverse chemicals, as part of the exposome, can be measured in a single analytical run by LC-MS/MS. A method was developed to measure >80 compounds in three different matrices (urine, serum and breast milk). These compounds include endocrine disruptors, plasticizers, perfluorinated alkylated substances, personal care product chemicals, disinfection by-products, components of cigarette smoke, phytochemicals, etc., sometimes present at very low concentrations. This method can be useful in Exposome-wide association studies. It is applied to the analysis of plasma samples in premature infants.

- Rationale for the selection of measured compounds is missing (insufficient details given in lines 116-17). Is the selection simply governed by opportunities, and a wish to include a few compounds from each type of exposure source? Will it be sufficient to establish this method or a derivative, as a reference method to measure the exposome?

- There is no mention of analytical blanks, whereas it is well known that a number of compounds analyzed here may also arise from contamination by leaching from lab tubes, pipes, etc. This was described in a previous publication by the authors (Preindl et al, 2019, Anal. Chem.). In the present manuscript the authors only refer to 'matrix blanks' in the supplementary information, which correspond to a commercial pooled plasma sample, but there is no mention of analytical blanks.

- In addition to such contaminations, other limitations commonly found when measuring non-persistent organic compounds should also be discussed (see e.g. Calafat et al., 2015, Environ Health Perspect). Figure 4 well illustrates the variability over time, a major limitation when measuring the exposome in EWAS studies.

- Result section on 'Estimation of daily toxicant intake of infants from breast milk measurements'. In this section, the authors develop models to evaluate risk associated with measured exposures. This discussion seems to go beyond the declared objectives of the study as summarized in the abstract (development of an analytical method to measure the chemical exposome).

- The 'high sensitivity of the method (...) generally outperformed previously published, more tailored multi-class assays' (line 370). The high sensitivity is emphasized. However, it is difficult to make one's opinion. They authors should provide data to compare their LOQs with LOQs of targeted assays in the literature, and ranges of concentrations as reported in previous HBM studies.

- In this study, about 80 compounds were measured in three different matrices. This is only a small fraction of the chemical exposome. The authors should discuss the feasibility to extend coverage of the exposome through the proposed method.

- The authors have included a few biotransformation metabolites of xenobiotics. They suggest using an enzymatic deconjugation to increase the number of metabolites detected (line 384). This could have been tested. This is certainly a valid proposition. However, it may make the distinction between true occurrence of contaminants in the biological matrix, and post sample collection contamination (during sample processing and analysis), more difficult.

- Line 212: why should the lower blood volumes be a logical explanation?

- Lines 226-245: This paragraph is more like discussion rather than the description of results. It should be moved to the discussion section.

- Line 250: Table S7 missing.

- Line 255: 'less dynamic' than what?

- Lines 285-286: 'For the first time, this study demonstrated that infants can be exposed to chemicals of emerging interest via breast milk'. Quite a number of papers have been published earlier on this topic, in particular for POPs.

- Line 342: 'fulfilling all criteria for at least one trace concentration level'. Not clear.

- Table S4, LOQ: add concentration units.

Reviewer #2 (Remarks to the Author):

The work by Jamnik and colleagues is intriguing. The group describes an analytical workflow for human biofluids that uses high-resolution mass spectrometry to measure a range of chemicals. Most notably they provide convincing evidence that it works in a rather unique population, namely neonates. Moreover, they demonstrate that the methods work in breast milk, which is a notoriously difficult biological matrix for mass spectrometry. I struggled through the manuscript until I realized that what threw me off was the title. The authors provide an overly broad title that doesn't highlight the uniqueness of their study. A title such as Next-generation biomonitoring of the early life exposome in neonatal and infant development OR Next-generation biomonitoring in breast milk and neonatal plasma would set the stage for the data that follow.

Figure 1 would seem better suited in a review paper. Figure 1 A shows the workflow and is appropriate. The tabular and figure display of the data in Table 1 and Figure 2 are sound. Similarly, the class-based description of the data in Figure 3 is very effective. The methods are described well and are state of the art.

Line 228 refers to potential sources of contamination in the neonatal ICU, but aren't those essentially components of the neonatal exposome? If the medical equipment causes chemicals to enter the infant's body it isn't contamination in the technical sense (as in an interference in the workflow), but rather an actual exposure in this vulnerable population. This is not to say that such interventions may provide life-saving medicines or nutrients, but rather there may be repercussions of the types of equipment that are used in the environment. Similarly, the authors mention contamination of the donated, pooled breast milk, but this is readily testable. Does the donated milk contain nicotine metabolites? This would seem an easy thing to measure.

The authors discuss the introduction of xenobiotics via breast milk, but should also explicitly mention the obvious benefits of antibodies, etc. Indeed, the authors should discuss the fact that formula is also going to contain a range of xenobiotics and it is important to make that comparison.

The concluding paragraph properly summarizes the work described. The authors should reframe the abstract and introduction to reflect the novelty of this study, i.e. the ability to measure a wide range of chemicals in biological matrixes that can reveal what chemicals infants are exposed to early in development. Yes, it is an ultra-sensitive approach, but it is the demonstration of its utility in neonatal plasma and breast milk that is most compelling. This work represents a major leap forward in understanding the interaction of the environment with early child development.

Reviewer #3 (Remarks to the Author):

First of all, I want to highlight that I had a very much interesting read regarding the submitted paper. This highly-timely paper is of high technical quality and reads well. The manuscript 'Next-generation biomonitoring of the chemical exposome in human biofluids' highlights the validation of an extended analytical LC-MS/MS technology of an existing methodology, including >80 exposome-related toxicants, applied to a subset of biological samples in a small cohort study.

This manuscript reads as the validation of an LC-MS/MS methodology applied on a set of biological samples, where the final conclusion is the exploratory verification of emerging toxicants in biological samples. The base of the method is an already published manuscript based on a single-group analysis method, and now extended to more toxicants. Therefore, in my opinion, this paper would better suit in Analytical Chemistry of another advanced analytical chemistry Journal. To date, there is also a special issue regarding exposome-biomonitoring in Environment International. Indeed, the 'exposome' field is a rapidly growing, emerging field, however, many points need to be unraveled and addressed. My main concern is that the authors promote this work as a tool to further explore associations with chronic diseases, however the question is 'are we looking to the right molecules'? I miss this section in the manuscript, are the compounds included the real validated biomarkers of exposure and/or effect? If you analyze >20 toxicants in a biological milk-matrix, what does this mean? Off course, this is exploratory research. The big deal is, if these toxicants will have an eventual effect. This is a bit of a misunderstanding of the authors, as they promote this research to use this tool to unravel these hypotheses, however, for most of the compounds we do not even know if we are looking to the correct biomarkers of exposure, effect of

even DNA-adducts (links to cancer). If this would have been extended in this paper with a clear outcome or association, I would definitely promote publication in Nature Communications.

In addition, the authors apply their methodology to blood-matrices, with a straightforward sample clean-up, however here I miss the use of an enzyme such as pronase (to account for albumin-bound compounds), and eventually the correction that needs to be verified for creatinine (urine) and hematocrit/albumin (blood). This is not reflected or discussed in the manuscript. Also, I could not verify how the authors use the 50 μL approach, when the method is validated for 200 μL of matrix. This is definitely not a straight-forward extrapolation, and required some in-depth research, applying these small volumes, definitely in terms of preconcentration.

The introduction reads well but is sometimes too general, and lacks some important detailed inclusions, such as:

- L.46: 'genomic research': what do you mean?
- L.55: 'unbiased data-driven': what do you mean?
- L.64: questionnaires, specify more, not only FFQ-questionnaires, also 24h recalls, also duplicate diet studies should be mentioned here.
- L.83: what is the definition of 'ultra-trace'?
- L.99: low-volume samples: what is the definition of low volume? In my opinion, 200 μL is still 'ok', less than 100 μL is low volume.
- Figure 1: the figure does not reflect what is included in the introduction (pathogenesis, precision medicine?)

The results section is well-elaborated:

- Why the choice to use the EC 657/2002 for validation, and not extend it with the 'Bioanalytical method validation of the US Department of Health and Human Services. Guidance for industry. 2018'?
- Not much is mentioned on the sample collection timing; the sampling of the breastmilk was done in a longitudinal way in 211 days, however what was the rationale of the 86 samples? What was the timing of the sampling: morning/evening/...? It is clear from recent papers that sample collection time is crucial, and more specifically that it is better to collect a 24h-matrix, instead of one-single-timepoint. How does this reflect (dietary) exposure? Could the authors make a preliminary screening on how the exposures were related to each other in a small model, besides the dynamics? As stated, this is interesting, however, only exploratory.
- Why only a high-exposure scenario was applied? To work with left-censored data, I assume different scenarios should be applied to account for sensitivity analyses.

The discussion reads well with focus on limitations of the study, along with the general exposure-research field, along with general recommendations for future ExWAS.

Online methods

- The use of plastic Eppendorf tubes: did the authors take into account possible adhesion of toxicants?
- Did the internal standards mixture account for all compounds? What was the rationale to use the specific concentrations of the mixture?
- Assessing linearity: is this the most optimal solution? Lack-of-fit test, taking weighing factors into account?

REVIEWER COMMENTS

Reviewer #1 (Remarks to the Author):

This work aims at the measurement of a panel of diverse xenobiotics in blood, urine and breast milk by liquid chromatography coupled to tandem mass spectrometry. It should be seen as a proof of principle study showing that highly diverse chemicals, as part of the exposome, can be measured in a single analytical run by LC-MS/MS. A method was developed to measure >80 compounds in three different matrices (urine, serum and breast milk). These compounds include endocrine disruptors, plasticizers, perfluorinated alkylated substances, personal care product chemicals, disinfection by-products, components of cigarette smoke, phytochemicals, etc., sometimes present at very low concentrations. This method can be useful in Exposome-wide association studies. It is applied to the analysis of plasma samples in premature infants.

- Rationale for the selection of measured compounds is missing (insufficient details given in lines 116-17). Is the selection simply governed by opportunities, and a wish to include a few compounds from each type of exposure source? Will it be sufficient to establish this method or a derivative, as a reference method to measure the exposome?

We would like to thank the reviewer for the generally positive feedback and the important questions/comments raised. For compound selection, it was a priority to include a very wide array of chemical classes and structural properties, but also to cover molecules with all kinds of toxicological modes of action. As discussed in the paper, a multitude of methods exist for specific chemical classes, but we intended to emphasize the technical feasibility to analyse structurally different compounds by this unique and novel generic assay at very low levels. The broad scope of the method is crucial to establish this workflow as a potential reference method for sequencing the exposome because it will be straight-forward to expand by other toxicants if needed. The specific compounds of the different chemical classes (well-known plastic components, PFAS, industrial chemicals, food processing by-products, myco- and phytoestrogens, personal care product ingredients, air pollutants and emerging disinfection by-products or phytotoxins) were then selected according to the following criteria:

- ✓ Feasibility of analysis: has effective ionisation and fragmentation utilizing reversed-phase-LC coupled to ESI - mass spectrometry been demonstrated before? Have previously reported detection limits of LC-MS/MS methods reached the concentrations that might occur in real-life biological samples?
- ✓ Toxic potential and literature reports on (suspected) impact on human health
- ✓ Abundance and relevance: which compounds best represent each chemical class/which compounds are often used as biomarker of exposure to the respective class?
- ✓ Metabolism/biotransformation: while the inclusion of the parent compound was of high importance (if feasible), the inclusion of additional relevant biotransformation products was considered; which phase I/II metabolites are primarily being produced and which ones might be of relevance to the mechanism of toxicity?
- ✓ Availability of authentic, NMR-confirmed reference standards: this was a main limiting factor for the addition of biotransformation products.

All of these factors were balanced accordingly in each case. Please find below some examples to showcase our selection process to enhance clarity:

- While the metabolites of nicotine are far from being the most toxic components of cigarette smoke, there is hardly a better specific proxy of cigarette smoke exposure.

- The direct analysis of PAHs using LC-ESI is hardly feasible because of inefficient ionisation. Therefore, only the main phase I metabolites were included as the additional hydroxy group helps with ionisation in negative mode. Pyrene and phenanthrene are abundant and toxic PAHs where good sensitivities were reached. While Benzo[a]pyrene is known as the most toxic member of this class, preliminary experiments showed that the additional ring systems did not enable efficient ionisation (using 1-hydroxy-benzo[a]pyrene).
- PhIP is known as a potent and abundant heterocyclic aromatic amine in cooked meat, hence it was included (bridging food and environmental toxicants). Since no relevant phase I metabolites (e.g. 4-hydroxy- and N²-hydroxy-PhIP), phase II metabolites (glucuronides, sulfates) or markers of toxicity (DNA adducts) were commercially available, we opted to test for the parent contaminant which we then actually identified for the first time in breast milk.

We elaborated further on the important topic of analyte selection in the paper, but still had to keep the information concise to keep a balanced word count. The adopted paragraph reads as follows:

“The choice of chemicals was based on the toxicological potential (i.e. is the compound of relevance in a health context?), real-life exposure levels (i.e. does this compound occur in quantities feasible to be determined?), suitability of the instrumental platform (i.e. is retention via reversed-phase LC and ionization via ESI-MS sufficient for sensitive analysis?) and representativeness for each class of toxicants to guarantee thorough monitoring of multiple adverse exposures (i.e. does this compound usually occur alongside other relevant chemicals of the same chemical class?).”
(Lines 115-120)

To answer the reviewer’s question, if ‘it will be sufficient to establish this method or a derivative, as a reference method to measure the exposome?’:

We believe this method holds a unique potential to be one reference method to assess the exposome. Given the broad scope of the exposome concept, there will never be only ‘one reference method’ but it will be important to keep in mind that we need other instrumental platforms too (e.g. GC-MS for toxicants that do not ionize well on LC-ESI-MS). Moreover, the selected panel of analytes should not be regarded as absolute and the expansion by other (emerging and maybe today unknown) contaminants and biotransformation products thereof constantly needs to be considered. Here, we are convinced that our newly-developed platform can be of especial value since it is generic in nature and thus enables continuous expansion with additional analytes for exposure- or direct effect monitoring. This is also highlighted in the discussion section.

- There is no mention of analytical blanks, whereas it is well known that a number of compounds analyzed here may also arise from contamination by leaching from lab tubes, pipes, etc. This was described in a previous publication by the authors (Preindl et al, 2019, Anal. Chem.). In the present manuscript the authors only refer to ‘matrix blanks’ in the supplementary information, which correspond to a commercial pooled plasma sample, but there is no mention of analytical blanks.

Thank you for bringing this up, it seems we did not describe in enough detail how we addressed this relevant problem in the initially submitted manuscript. Of course we included analytical blanks in our work. Analytical blanks (extracted water) were not only applied during method validation (see Online Methods – named as “system blanks”), but also in each individual measurement sequence as multiple analytes were either leachables of the LC system or introduced during sample preparation to some degree (among them mostly plasticizers: BPA, BPS, MBP, MEHP, benzyl butyl-/dibutyl phthalate, n-butylbenzenesulfonamide, nonylphenol).

These minor background contaminations were thoroughly quantitated utilizing solvent calibration and subsequently subtracted from the calculated concentrations in biological samples during each measurement sequence. We now further eluded on this topic in the Online Methods section (Quantitation, data evaluation and figures) as follows:

“Importantly, results of unknown samples as well as results of method validation were additionally corrected by the system contamination with certain compounds that most likely occurred as extractables and/or leachables from the used laboratory tubes during sample preparation and/or the LC instrumentation itself (BPA, BPS, MBP, MEHP, benzyl butyl-/dibutyl phthalate, n-butylbenzenesulfonamide, nonylphenol). This was done by applying triplicate analytical blanks (extracting water) and determining the extent of contamination via solvent regression.” (Lines 24-629)

- In addition to such contaminations, other limitations commonly found when measuring non-persistent organic compounds should also be discussed (see e.g. Calafat et al., 2015, *Environ Health Perspect*). Figure 4 well illustrates the variability over time, a major limitation when measuring the exposome in EWAS studies.

We thank the referee for this valuable comment and discussed these issues in a new paragraph as follows:

“The readily accessible bio-fluids urine and blood are the most commonly used matrices in established and automated sample preparation strategies⁴⁹. Independent of the route of exposure, ingested compounds are anticipated in blood because this bio-fluid is in steady-state contact with most human tissues. Therefore, it may be considered as the best-suited matrix to characterize the chemical exposome⁵⁰. In parallel, complementary urinary analyses can shed light on short-lived compounds or excretory metabolites. Colorimetric assays to determine creatinine in urine and albumin in blood may be implemented to report xenobiotic levels in relation to the respective bio-fluid concentrations (see Supplementary Table S7).

However, it is vital to critically question which toxicity-related questions can be answered by the data created using such exposome-scale HBM methods. This clearly depends on the investigated biological matrix and the chemical properties of a specific toxicant. Currently, our assay encompasses predominantly parent compounds, and a few primary metabolites, of representative xenobiotics of multiple chemical classes. Blood may be a suitable matrix to assess chronic contamination with persistent organic pollutants (e.g. PFAS), while for immediately metabolized and excreted compounds such as phthalates, bisphenols or parabens, the parent compounds are more of a proxy of acute rather than chronic ingestion⁵¹. For such chemicals that do not feature any persistent metabolites, longitudinal urinary analysis of the predominant phase-II metabolites, either by a direct assessment or by the application of deconjugation, may be better suited to assess chronic exposures.” (Lines 398-415)

- Result section on ‘Estimation of daily toxicant intake of infants from breast milk measurements’. In this section, the authors develop models to evaluate risk associated with measured exposures. This discussion seems to go beyond the declared objectives of the study as summarized in the abstract (development of an analytical method to measure the chemical exposome).

Thank you for this remark. We now mention this additional objective in the abstract for the sake of completeness:

“Based on the generated data, a preliminary estimation of daily toxicant intake via breast milk is conducted.” (Lines 26-27)

In fact, we believe that this is an important translational aspect of the manuscript. In contrast to just reporting exposure data, we showcase that these results can be used for a wide variety of toxicants by regulatory bodies and believe that this angle of the work might be appealing to toxicologists and risk assessors (beyond analytical chemists, exposure scientists and environmental health professionals).

- The 'high sensitivity of the method (...) generally outperformed previously published, more tailored multi-class assays' (line 370). The high sensitivity is emphasized. However, it is difficult to make one's opinion. They authors should provide data to compare their LOQs with LOQs of targeted assays in the literature, and ranges of concentrations as reported in previous HBM studies.

Indeed, a major breakthrough of this work is the achievement of trace-level sensitivities for multiple, structurally unrelated chemical classes in a single assay. Please find below a direct comparison with other recently published multi-class HBM methods. We added this table to the SI material (Table S6) and refer to it within the text of the main paper.

Table S6: Sensitivity comparison between the new multi-class assay and published multi-class methods.

Compound	Matrix	LOQ [ng mL ⁻¹]	LOQ [ng mL ⁻¹] ¹ (published method)	Published method	Comment
Bisphenol A (BPA)	Urine*	0.21	0.33*	Heffernan et al, Talanta 151, 224-233, (2016) ²	19 compounds total, 2 chemical classes
Bisphenol AF (BPAF)		0.038	0.017*		
Bisphenol B (BPB)		0.022	0.87*		
Bisphenol F (BPF)		0.064	1.3*		
Bisphenol S (BPS)		0.036	0.22*		
Mono-n-butyl phthalate (MBP)		0.34	0.17*		
Mono-2-ethylhexyl phthalate (MEHP)		0.047	0.067*		
Bisphenol A (BPA)	Urine	0.21	0.1	Rocha et al, Talanta 183, 94-101, (2018) ³	21 compounds total, 3 chemical classes
Bisphenol AF (BPAF)		0.038	0.04		
Bisphenol F (BPF)		0.064	0.25		
Bisphenol S (BPS)		0.036	0.07		
Benzophenone 1		0.0054	0.1		
Benzophenone 2		0.018	0.07		
Benzylparaben		0.00055	0.1		
Butylparaben		0.0072	0.1		
Ethylparaben		0.0039	0.05		
Methylparaben		0.048	0.05		
Propylparaben		0.011	0.1		
Triclosan		0.035	0.5		
Bisphenol A (BPA)		Serum/ Plasma ⁺	0.57		
Bisphenol AF (BPAF)	0.075		0.15		
Bisphenol F (BPF)	0.068		0.044		
Bisphenol S (BPS)	0.0086		0.055		
Benzylparaben	0.0011		0.20		
Butylparaben	0.0096		0.13		
Ethylparaben	0.019		0.15		
Methylparaben	0.035		0.17		
Propylparaben	0.0081		0.17		
Estrone (E1)	0.0085		0.11		
Estradiol (E2)	0.076		0.006		
Estriol (E3)	0.058		0.009		
Bisphenol A (BPA)	Urine		0.21	0.2	Bocato et al, Environ Res 189 (2020) ⁵
Bisphenol AF (BPAF)		0.038	0.2		
Bisphenol F (BPF)		0.064	0.15		
Bisphenol S (BPS)		0.036	0.14		
Benzophenone 1		0.0054	0.05		
Benzophenone 2		0.018	0.1		
Benzylparaben		0.00055	0.08		
Butylparaben		0.0072	0.03		
Ethylparaben		0.0039	0.1		
Methylparaben		0.048	0.15		
Propylparaben		0.011	0.02		
Triclosan		0.035	0.5		
Bisphenol A (BPA)		Urine	0.21	0.2	
Benzylparaben	0.00055		0.05		
Butylparaben	0.0072		0.04		
Ethylparaben	0.0039		0.04		
Methylparaben	0.048		0.04		
Propylparaben	0.011		0.04		
Triclosan	0.035		0.12		

*Heffernan et al. only reported the limit of detection as a signal to noise ratio of three. The limit of quantification was calculated by multiplication with 10/3 to estimate a signal to noise ratio of 10.

+ As no published multi-class assay for serum was identified, the sensitivity of our assay in serum could only be compared to the sensitivity of the assay by Kolatorova et al. which was developed for plasma

Kindly note that the definition of ‘multi-class assay’ demands the inclusion of multiple structurally different chemical classes of toxicants. Importantly, the literature concerning such broad methods is still rather scarce and mostly limited to ~20 synthetic compounds (parabens, bisphenols, benzophenones). Furthermore, we are not aware of any multi-class method that has been applied to human breast milk to date, so no comprehensive comparison was feasible.

- In this study, about 80 compounds were measured in three different matrices. This is only a small fraction of the chemical exposome. The authors should discuss the feasibility to extend coverage of the exposome through the proposed method.

We fully agree with the reviewer and see this work as an important, but certainly not final step for assessing chemical exposure holistically. As mentioned above, it was a primary focus to keep the sample preparation and chromatographic separation as generic as possible to allow for a further extension of the included markers of exposure without much additional effort and without reaching technical limitations of the instrumentation. The most important element for this is the generic sample preparation strategy. As requested, we are now discussing this aspect in more depth in the Discussion section as follows:

“These issues highlight an important advantage of the presented approach, namely its extendable design. At the technical level, the LC-MS/MS system is still far from working at full capacity, as it still holds the potential of the inclusion of hundreds of additional mass spectrometric transitions. With future advances concerning the knowledge of emerging toxins or relevant biotransformation products as well as the availability of appropriate analytical standards, each compound category might be further expanded with additional markers of exposure or direct biomarkers of biological effect. While measuring the ‘entire exposome’ by a single analytical method or platform will not be feasible, we see this approach as an ambitious attempt to cover a vast chemical space that can be further expanded.” (Lines 426-433)

- The authors have included a few biotransformation metabolites of xenobiotics. They suggest using an enzymatic deconjugation to increase the number of metabolites detected (line 384). This could have been tested. This is certainly a valid proposition. However, it may make the distinction between true occurrence of contaminants in the biological matrix, and post sample collection contamination (during sample processing and analysis), more difficult.

We fully agree with the referee and have performed extensive additional experiments to test the potential of deconjugation comprehensively. For this purpose, we treated all of the before analysed breast milk samples with β -glucuronidase/arylsulfatase mixture from *Helix pomatia*. Moreover, we treated and re-analyzed some of the plasma samples from the premature infants to get preliminary information on conjugation patterns. Due to ethical constraints during the sampling process (you must not draw more than a couple of blood drops from these susceptible neonates), we only had enough material to do these re-analyses for three samples where still some plasma (40 μ L) was left from the original analyses.

The results for all individual breast milk samples after enzymatic hydrolysis with beta-glucuronidase/arylsulfatase are now reported in Table S10 and a comparison between non-digested (Table S9) and digested samples is given in Table S11. This comparison is also inserted below to highlight the effect of enzymatic deconjugation:

Table S11 (please see SI for additional comments/explanations and better resolution): Comparison of analyte concentrations in breast milk samples before and after treatment with beta-glucuronidase and sulfatase (*Helix pomatia*).

Compound	Longitudinal breast milk (n=86)			Enzymatically treated longitudinal breast milk (n=86)		
	Pos. samples	Min. [ng mL ⁻¹]	Max. [ng mL ⁻¹]	Pos. samples	Min. [ng mL ⁻¹]	Max. [ng mL ⁻¹]
Plasticizer/Plastic Components						
Bisphenol A (BPA)	2	<0.40	<0.40	4	0.26	5.6
Bisphenol AF (BPAF)		-		1	0.047	0.047
Bisphenol F (BPF)	1	<0.028	<0.028		-	
Bisphenol S (BPS)	57	<0.0078	0.05	25	<0.0078	0.086
Mono-n-butyl phthalate (MBP)	79	<0.16	6.4	8	<0.16	17
N-butylbenzenesulfonamide	86	3.4	14	86	<1.5	2.9
Perfluorinated Alkylated Substances						
Perfluorooctanoic acid (PFOA)	79	<0.092	<0.092		-	
Perfluorooctanesulfonic acid (PFOS)	74	<0.016	0.048		-	
Industrial Side Products and Pesticides						
2-naphthol	86	0.21	1.1	79	0.88	4.6
Prochloraz	1	0.082	0.082		-	
Endogenous Estrogens						
Estriol (E3)		-		1	0.47	0.47
Phytoestrogens and Metabolites						
8-prenylnaringenin	17	<0.015	0.23	57	<0.015	0.24
Coumestrol		-			-	
Daidzein	44	<0.0083	0.14	4	0.31	2.2
Enterodiol	12	0.0079	0.025	13	<0.0014	0.24
Enterolactone	3	<0.17	<0.17	85	<0.17	4.2
Genistein		-		10	0.24	1.2
Glycitein	23	<0.0023	0.011		-	
Isoxanthohumol	14	<0.0095	0.078		-	
Resveratrol	1	<0.30	<0.30		-	
Xanthohumol	3	<0.22	<0.22	65	0.59	21
Mycostrogens and Metabolites						
Alternariol	1	0.51	0.51	10	<0.076	2.3
Personal Care Product Ingredients, Pharmaceuticals and Metabolites						
Benzophenone 1	86	<0.012	0.039	5	<0.012	0.21
Benzophenone 2	2	<0.0096	0.02	5	<0.0096	0.052
Butylparaben (BP)	14	<0.0099	0.036	10	<0.0099	0.056
Ethylparaben (EP)	86	<0.041	0.13	75	<0.041	1.2
Methylparaben (MP)	86	0.069	23	86	<0.028	11
Propylparaben (PP)	86	0.033	16	86	<0.01	4.9
Triclosan		-		1	<1	<1
Phytotoxins						
Anisodamine	9	<0.0010	0.031		-	
Jacobine-N-oxide	3	<0.035	0.11		-	
Riddelliin-N-oxide	1	<0.021	<0.021		-	
Scopolamine	4	<0.00014	0.0004		-	
Food Processing By-Products						
PhIP	2	<0.0042	<0.0042		-	

The results for the three neonate plasma samples after enzymatic hydrolysis with beta-glucuronidase/arylsulfatase are now reported in Table S7 together with a comparison to the samples without enzymatic treatment. A selection of representative analytes was evaluated to examine if concentrations increase upon enzymatic hydrolysis.

The additional experiments confirmed our assumption that for many toxicants increased concentrations can be found (e.g. most phytoestrogens). However, as also mentioned by the referee, applying this treatment is not always straight-forward. The reasons were as follows: (1) Contamination of the crude snail extract with the xenobiotic and thus a high background signal, (2) retention time shifts out of the sMRM window caused by the pH of the utilized buffer, (3) a general higher background noise in samples, which we could also see in other assays after enzymatic treatment in the past, that reduces the method's sensitivity. Consequently, some analytes could not be assessed at all after treatment in breast milk (e.g. phytotoxins and PhIP).

We conclude that deconjugation can add additional information to a data set but that it is not straight-forward to be applied in a multi-class assay covering a broad chemical space. The additional experiments certainly underlined this fact and considerably added to the general impact of this work and we would like to thank the referee to ask for these additions.

Ideally, one would always measure a sample twice, once with and once without enzymatic deconjugation. This way it would be feasible to assess the non-conjugated native toxin (active form in most cases) and the conjugated (detoxified) portion. However, as we saw in the neonate cohort, this is not always feasible because of extremely low volumes that might be sampled e.g. during early life which are only sufficient for a one-time analysis. The best case would be a method that can directly measure detoxified as well as active species and even direct biomarkers of effect (e.g. DNA-adducts). However, as we allude to on several occasions, the availability of analytical standards is still a main barrier and we did include some biotransformation products for which NMR-confirmed standards were available.

The experimental procedures were added to the 'Online Methods' section and the additional supplementary tables mentioned in the text of the revised manuscript. The results are now also part of the *Discussion* section:

*"As the volumes of the available materials were mostly sufficient for only one sample preparation, an enzymatic deconjugation of the infant plasma samples was merely tested for a few samples in this work. Moreover, all breast milk samples were enzymatically treated and re-measured. In addition to shifts in retention times, caused by the pH of the buffer, the introduction of a crude enzyme extract (β -glucuronidase and sulfatase from *Helix pomatia*) severely increased baseline noise and contamination for many analytes in both biological matrices. As the method was analytically validated without enzymatic treatment, this partially resulted in selectivity and sensitivity issues for several analytes. Nonetheless, the experiments confirmed the assumption that increased toxicant concentrations may be unmasked for compounds where no severe analytical interference was observed. In future studies, this approach has the potential to increase the number and concentrations of the xenobiotics detected, most notably in urine, following thorough re-validation. The use of proteolytic enzymes (e.g. pronase) in future work might be considered if highly electrophilic molecules with an affinity for albumin/protein adducts are in focus."* (Lines 385-397)

- Line 212: why should the lower blood volumes be a logical explanation?

We thank the referee for this comment and have re-written this passage as follows:

"Comparison of these concentrations with adult control samples revealed that the contamination of the infant samples with certain chemicals was substantially higher in many subjects. For example, 11 ng mL⁻¹ BPA, 12 ng mL⁻¹ PFOA, 17 ng mL⁻¹, 2-naphthol, 9.4 ng mL⁻¹, 4-tert-octylphenol, 0.13 ng mL⁻¹, ethylparaben, 15 ng mL⁻¹, methylparaben, and 2.9 ng mL⁻¹ propylparaben were detected in one infant. These values are all higher than the concentrations determined for both adult controls. Clearly, the adults were not in contact with the same potential intensive care unit (ICU)-related exposure sources (see Discussion) as the baby cohort. However, it is also known that infants have a reduced drug detoxification capability³¹. Consequently, with a lower proportion of conjugated species there might be a higher level of more potent xenobiotics in most of the participants in the cohort." (Lines 212-221)

- Lines 226-245: This paragraph is more like discussion rather than the description of results. It should be moved to the discussion section.

We appreciate the comment and agree; the paragraph was moved as suggested.

- Line 250: Table S7 missing.

Additional Supplementary Tables were initially submitted in .xls format (in separate sheets of the same file) since they were too big to be reasonably depicted as a .doc file. In case these tables were not provided in the initial review, we apologize and would kindly ask the editorial office to make them assessable easily. In fact, these tables are very important from our point of view as they should enable secondary use by fellow researches.

- Line 255: 'less dynamic' than what?

This comparison was relating to the differences in time-exposure patterns in breast milk between natural- and synthetic xenobiotics. The statement was adapted for clarity:

“Detected levels spanned a wide concentration range over the observed period (i.e. < 0.0083 ng mL⁻¹ – 0.14 ng mL⁻¹ for daidzein). In most cases, contamination with synthetic chemicals such as parabens (Figure 4 a and b), PFAS, and phthalates was less dynamic when compared to xenobiotics of natural origin. These observations are consistent with the expected chemical properties (i.e., toxicokinetic half-lives, adduct binding in plasma) and route of exposure. Phytoestrogen exposure is dependent on nutritional intake, which typically changes daily as is exemplified by daidzein depicted in Figure 4 d. Paraben exposure, however, is caused by consumer care product usage, which is likely not as variable as nutritional behaviour.” (Lines 238-246)

- Lines 285-286: 'For the first time, this study demonstrated that infants can be exposed to chemicals of emerging interest via breast milk'. Quite a number of papers have been published earlier on this topic, in particular for POPs.

Thank you for highlighting this fact, which is of course true. The statement was supposed to be framed in relation to the first detection of phytotoxins exclusively. It was re-phrased for clarification:

“Phytotoxin contamination was apparent throughout the sampling period as the individual was known to regularly consume herbal tea mixtures that are intended to promote milk production. For the first time, this study demonstrated that infants can be exposed to these chemicals of emerging interest via breast milk if the mother regularly consumes certain foodstuffs or drinks.” (Lines 274-278)

- Line 342: 'fulfilling all criteria for at least one trace concentration level'. Not clear.

Thank you for bringing this to our attention. The statement was re-phrased for clarification:

“For urine and serum, the method yielded highly-convincing validated results with more than 75% of the chemicals included fulfilling all criteria for at least one trace concentration level that was applied for the spiking experiments.” (Lines 332-335)

- Table S4, LOQ: add concentration units.

Thank you for the comment – done.

Reviewer #2 (Remarks to the Author):

The work by Jamnik and colleagues is intriguing. The group describes an analytical workflow for human biofluids that uses high-resolution mass spectrometry to measure a range of chemicals. Most notably they provide convincing evidence that it works in a rather unique population, namely neonates. Moreover, they demonstrate that the methods work in breast milk, which is a notoriously difficult biological matrix for mass spectrometry. I struggled through the manuscript until I realized that what threw me off was the title. The authors provide an overly broad title that doesn't highlight the uniqueness of their study. A title such as Next-generation biomonitoring of the early life exposome in neonatal and infant development OR Next-generation biomonitoring in breast milk and neonatal plasma would set the stage for the data that follow.

We thank the reviewer very much for these kind words which are truly encouraging. We are glad that it is acknowledged that we selected challenging biological matrices and a population that is unique and not straight-forward to assess.

We adapted the title as suggested to be more specific and are grateful for the perspective offered by this referee. It now reads '*Next-generation biomonitoring of the early-life chemical exposome in neonatal and infant development*'.

Figure 1 would seem better suited in a review paper. Figure 1 A shows the workflow and is appropriate. The tabular and figure display of the data in Table 1 and Figure 2 are sound. Similarly, the class-based description of the data in Figure 3 is very effective. The methods are described well and are state of the art.

We agree that Figure 1 would also be well-suited in a review paper. However, we feel it is of value for setting the stage for the emerging paradigm of the exposome as the concept is still not main stream. Moreover, we intended to enhance the attention of a broad auditorium, especially medical professionals and epidemiologists, to this important but under-represented and under-researched field. Therefore, we would prefer to keep Figure 1 as is. However, of course we would be open to remove it if the referee or the editors still believes it is of limited value.

Line 228 refers to potential sources of contamination in the neonatal ICU, but aren't those essentially components of the neonatal exposome? If the medical equipment causes chemicals to enter the infant's body it isn't contamination in the technical sense (as in an interference in the workflow), but rather an actual exposure in this vulnerable population. This is not to say that such interventions may provide life-saving medicines or nutrients, but rather there may be repercussions of the types of equipment that are used in the environment. Similarly, the authors mention contamination of the donated, pooled breast milk, but this is readily testable. Does the donated milk contain nicotine metabolites? This would seem an easy thing to measure.

We highly appreciate this insightful comment which is absolutely valid. We used the term "contamination" with less technical attention than the referee did in this context. In a technical sense, it would not really be an "unwanted presence of toxic substances" if e.g. parabens were purposely injected as part of a drug's formulation (excipients). Our aim was to simply conjure neutral hypotheses relating to the source of the chemicals we detected without discrediting the potential life-saving effects of these interventions, as mentioned.

To better clarify this, we opted to remove the word "contamination" in these statements:

"Potential sources of exposure to plasticizers include contact with medical equipment such as ventilators, feeding tubes and infusion pumps in the neonatal intensive care unit. High paraben

levels may stem from the injection of pharmaceuticals that include them as excipients or preservatives⁵².” (Lines 438-441)

Concerning the nicotine metabolites in the breast milk: This is an interesting issue that we did not discuss sufficiently in the initially submitted version of the paper. Unfortunately, we do not possess the breast milk that has been used by the paediatric ICU to feed the neonates, so we were not able to verify the source of exposure at the time of sampling.

However, during analytical method development and validation using the breast milk pool that was provided by the Semmelweis Women’s Clinic of Vienna (a different clinic), it was demonstrated to be contaminated with cotinine and trans-3-hydroxy cotinine. We asked the person in charge of the milk bank and confirmed that only non-smoking mothers with excessive milk production are permitted as milk donors. We concluded that (the very low concentrations of) nicotine metabolites were most likely caused by passive smoking of one of the milk donating mothers and was only traceable since our method is highly sensitive. To further examine this point, we will perform expanded analyses of donated breast milk in the future.

Side note: This ultimate sensitivity was also confirmed in a recent biomonitoring study among Austrian primary school kids (aged 6 - 10). Here, we saw that even some of those young kids had detectable levels of nicotine metabolites in their urine which is most likely associated with passive smoking as well (*unpublished*).

Again, we would like to acknowledge this comment and adapted the paragraph to include this finding as follows:

“The detection of cotinine in 20 individuals from the cohort points towards the possibility of systemic passive exposure to smoke. At first glance, this was a particularly unexpected outcome. Although it was not possible to accurately quantitate cotinine using matrix-matched calibration (high contamination of the commercially-available pooled human serum used for calibration purposes); by applying solvent calibration, levels <0.5 ng mL⁻¹ were estimated for all samples. This is consistent with previous findings of mean cotinine levels in non-smokers as a result of everyday exposure to passive smoking⁵³. The contamination of donated, pooled breast milk that is routinely used as a supplement in the clinic may also be a source of this ubiquitous contamination, although active smokers are typically excluded as milk donors. This hypothesis is reinforced by the identification of both nicotine metabolites cotinine and trans-3-hydroxy cotinine in the breast milk that was applied for matrix-matched calibration (Supplementary Figure S1). However, this breast milk pool was provided by a different clinic and was not the pool that had been fed to the neonate cohort (see Online Methods).” (Lines 441-453)

The authors discuss the introduction of xenobiotics via breast milk, but should also explicitly mention the obvious benefits of antibodies, etc. Indeed, the authors should discuss the fact that formula is also going to contain a range of xenobiotics and it is important to make that comparison.

We fully agree and certainly do not intend to weaken the positive health effects of breast milk. In fact, exposure derived from commercial infant food/milk substitutes typically contain higher levels of chemical contaminants (besides their lack of antibodies, appropriate oligosaccharides etc.). We proved this point for mycotoxins recently in two publications that should make our view on this crucial topic very clear:

- (1) Braun et al. (2021): Natural contaminants in infant food: The case of regulated and emerging mycotoxins, Food Control 123, 107676 (<https://doi.org/10.1016/j.foodcont.2020.107676>).

(2) Ezekiel et al. (2020): Comprehensive mycotoxin exposure biomonitoring in breastfed and non-exclusively breastfed Nigerian children, MedRxiv preprint (<https://www.medrxiv.org/content/10.1101/2020.05.28.20115055v2>)

To transport this message in a better way, we added to the text as follows:

“While we focused on the potential contribution of breast milk towards the total xenobiotic burden of a neonate in this work, it must be clearly stated that low-level chemical exposure through breastmilk shall not be a factor to reduce or avoid breastfeeding. We recently demonstrated that alternatively consumed infant formula likely leads to higher mycotoxin exposure than the consumption of breast milk⁴⁴. This might be also true for other toxicant classes. Most importantly, the benefit of antibodies, growth factors and bioactive compounds including oligosaccharides in breast milk must be explicitly mentioned⁵⁵.” (Lines 482-488)

The concluding paragraph properly summarizes the work described. The authors should reframe the abstract and introduction to reflect the novelty of this study, i.e. the ability to measure a wide range of chemicals in biological matrixes that can reveal what chemicals infants are exposed to early in development. Yes, it is an ultra-sensitive approach, but it is the demonstration of its utility in neonatal plasma and breast milk that is most compelling. This work represents a major leap forward in understanding the interaction of the environment with early child development.

We thank the referee for this comment. As mentioned, we changed the title of the manuscript to better reflect the uniqueness of the work and also modified other parts of the paper to better transport the message that we now have a unique assay to better understand environmental exposure during early life.

Reviewer #3 (Remarks to the Author):

First of all, I want to highlight that I had a very much interesting read regarding the submitted paper. This highly-timely paper is of high technical quality and reads well. The manuscript 'Next-generation biomonitoring of the chemical exposome in human biofluids' highlights the validation of an extended analytical LC-MS/MS technology of an existing methodology, including >80 exposome-related toxicants, applied to a subset of biological samples in a small cohort study.

This manuscript reads as the validation of an LC-MS/MS methodology applied on a set of biological samples, where the final conclusion is the exploratory verification of emerging toxicants in biological samples. The base of the method is an already published manuscript based on a single-group analysis method, and now extended to more toxicants. Therefore, in my opinion, this paper would better suit in Analytical Chemistry of another advanced analytical chemistry Journal. To date, there is also a special issue regarding exposome-biomonitoring in Environment International.

Thank you very much for your thorough evaluation of our work and the kind words. This is highly appreciated as the development and validation of this unique method was a year-long effort.

We agree that there is a lot of analytical novelty in this piece and are also well aware of the current special issue on exposomics in *Environment International* (the senior author of the paper at hand is one of the editors who proposed the topic). However, we intentionally wrote this paper with a broad scope to appeal to a broad audience and to promote the emerging exposome concept beyond traditional discipline borders. Particularly, we intend to highlight the technical feasibility of holistic, low-level (early-life) exposure assessments to a wide readership including medical professionals, biologists, toxicologists, and policy makers. Only by strengthening cooperation between our disciplines will we be able to unravel and address the complexity that the exposome represents when it comes to the evaluation of health effects. We feel that the high level of novelty with regards to both, technical developments and insights into a highly vulnerable population (extremely premature neonates) justifies this approach and are encouraged by the generally very positive response of all three referees and the editorial team.

Indeed, the 'exposome' field is a rapidly growing, emerging field, however, many points need to be unraveled and addressed. My main concern is that the authors promote this work as a tool to further explore associations with chronic diseases, however the question is 'are we looking to the right molecules'? I miss this section in the manuscript, are the compounds included the real validated biomarkers of exposure and/or effect? If you analyze >20 toxicants in a biological milk-matrix, what does this mean? Off course, this is exploratory research. The big deal is, if these toxicants will have an eventual effect. This is a bit of a misunderstanding of the authors, as they promote this research to use this tool to unravel these hypotheses, however, for most of the compounds we do not even know if we are looking to the correct biomarkers of exposure, effect of even DNA-adducts (links to cancer). If this would have been extended in this paper with a clear outcome or association, I would definitely promote publication in Nature Communications.

We appreciate this comment and welcome the question 'are we looking to the right molecules?'. We believe that this question was partially touched in our reply to referee #1 (selection of toxicants included in the assay). It is also linked to the highly relevant discussion of exposure monitoring vs (biological-) effect monitoring. Exposure monitoring cannot directly answer the question concerning actual toxic (mixture) effects. It can, however, improve our understanding and help us find these answers through the generation of associations if enough statistical power is provided. It is correct that we need to state more clearly that this work is an advanced exposure monitoring tool to find associations with disease currently. However, since our workflow is scalable it seems plausible to expand it by the inclusion of specific biomarkers of effect such as

DNA-adducts. We have done this in our other, more tailored, single toxicant class assays (e.g. by directly quantifying the guanine-adduct of aflatoxin as effect marker alongside multiple aflatoxin exposure markers; Braun et al. 2020 (<https://doi.org/10.3389/fchem.2020.00423>)) and will consider this valid proposition for our next expansion.

Of course, the inclusion of additional markers of toxicity (e.g. PAH/acrylamide/PhIP/aristolochic acid-DNA adducts, acrylamide/pyrrolizidine alkaloid-protein/DNA adducts, ...) would improve the direct effect monitoring-side of our approach. Effect monitoring is a topic we are highly interested in and we are actively connecting to groups that are specializing in this topic (e.g. Balboa lab at the Mansonic Cancer Center, Minnesota, Sturla lab at ETH Zurich). However, similarly than with phase-II metabolites or activated metabolites that are “closer” to the direct adverse effect than the parent compounds, we are currently still strongly limited by the availability of authentic analytical standards. Fortunately, the method’s generic nature enables an easy inclusion of new compounds once available or of emerging interest as pointed out above. This was a main intention behind the method’s setup, as mentioned multiple times in the manuscript.

As you commented, for many chemicals we do not know specific markers of toxicity to date. This is especially complex e.g. for EDCs, as many of these exposures might act in a combinatorial manner via multiple avenues in the endocrine system. For the currently included compounds, we know that they carry the mechanistic potential to cause adverse health effects. As stated in the manuscript, they were chosen according to their abundance relating to their specific group, toxicity and thus relevance, and feasibility of analysis. By including the non-detoxified parent compound as a highest priority, we cover all the potential mechanisms of actions that may lead to disease. With exposome-wide association studies, we may be able to put actual numbers to such potential relationships in the future (this is a long-term aim of our research).

We now discuss the raised issues in an additional paragraph in the *Discussion* section of the revised paper:

“However, it is vital to critically question which toxicity-related questions can be answered by the data created using such exposome-scale HBM methods. This clearly depends on the investigated biological matrix and the chemical properties of a specific toxicant. Currently, our assay encompasses predominantly parent compounds, and a few primary metabolites, of representative xenobiotics of multiple chemical classes. Blood may be a suitable matrix to assess chronic contamination with persistent organic pollutants (e.g. PFAS), while for immediately metabolized and excreted compounds such as phthalates, bisphenols or parabens, the parent compounds are more of a proxy of acute rather than chronic ingestion⁵¹. For such chemicals that do not feature any persistent metabolites, longitudinal urinary analysis of the predominant phase-II metabolites, either by a direct assessment or by the application of deconjugation, may be better suited to assess chronic exposures. Moreover, the method does not yet include direct markers of biological effect. Consequently, we believe this work is of prime relevance in the area of exposure monitoring, rather than biological effect monitoring. This distinction needs to be stated clearly, as the former may be used to only find associations between disease development and exposure patterns. We encourage the creation of sufficiently large data sets by exposome-scale investigations to increase statistical power. Biomarkers of biological effect directly assess the adverse effect of a xenobiotic. For many chemicals no specific biomarkers of effect have been defined as their multidimensional mechanisms (e.g. EDCs acting on multiple fronts in the hormonal system) are still not fully elucidated. For mechanistically simpler, adduct-forming toxicants (e.g. acrylamide, PAHs, PhIP or phytotoxins) the availability of analytical standards severely limits the inclusion of DNA- or protein conjugates to date.

These issues highlight an important advantage of the presented approach, namely its extendable design. At the technical level, the LC-MS/MS system is still far from working at full capacity, as it

still holds the potential of the inclusion of hundreds of additional mass spectrometric transitions. With future advances concerning the knowledge of emerging toxins or relevant biotransformation products as well as the availability of appropriate analytical standards, each compound category might be further expanded with additional markers of exposure or direct biomarkers of biological effect. While measuring the 'entire exposome' by a single analytical method or platform will not be feasible, we see this approach as an ambitious attempt to cover a vast chemical space that can be further expanded." (Lines 406-433)

Regarding the question on outcomes/associations: Working with samples from extreme premature infants (<1 kg birth weight and <28 weeks of gestational age) is extremely challenging on many levels: (1) Only small cohorts can be obtained as even large clinics do not have many patients and of those that are born alive the mortality within the first weeks/months of live is comparably high. (2) Ethics and sample volume: Preterm infants represent a heterogeneous population with multifarious needs at different stages of life (mechanical ventilation, enteral feeding, prenatal & prematurity-related complication etc.). It is ethically critical to obtain blood samples from such individuals and (if granted and feasible), the obtained volumes are very limited. (3) In longitudinal settings it is typically not feasible to obtain samples at all time points for all infants.

Here, we focused our research on the feasibility to work with such precious low-volume samples to lay the foundation for larger association studies of this highly vulnerable patient cohort in the future. We fully agree that for the most part we currently don't fully understand which compounds actually exhibit potential effects on patients. During an intensive care stay over several months, these high-risk patients are confronted with various compounds at different stages of development. We believe that close longitudinal sampling is necessary to unravel so-far unknown associations with health- and developmental outcomes in this cohort. This will be a year-long effort that requires significant long-term funding. Linking detected compounds with prematurity-related outcomes (bronchopulmonary dysplasia, retinopathy of prematurity, neurodevelopment etc.) at a single time point in a relatively small heterogeneous patient cohort might lead to wrong associations. Therefore, we did not associate clinical outcome data with the obtained compound concentrations at this point.

In addition, the authors apply their methodology to blood-matrices, with a straightforward sample clean-up, however here I miss the use of an enzyme such as pronase (to account for albumin-bound compounds), and eventually the correction that needs to be verified for creatinine (urine) and hematocrit/albumin (blood). This is not reflected or discussed in the manuscript.

Thank you very much for this comment. We now mention the possibilities of concentration-corrections via albumin/creatinine by usage of additional colorimetric assays as suggested and also added the corresponding hematocrit/albumin data to the manuscript:

"Independent of the route of exposure, ingested compounds are anticipated in blood because this bio-fluid is in steady-state contact with most human tissues. Therefore, it may be considered as the best-suited matrix to characterize the chemical exposome⁵⁰. In parallel, complementary urinary analyses can shed light on short-lived compounds or excretory metabolites. Colorimetric assays to determine creatinine in urine and albumin in blood may be implemented to report xenobiotic levels in relation to the respective bio-fluid concentrations (see Supplementary Table S7)." (Lines 399-405)

"Detailed results are reported in Supplementary Table S7 and include hematocrit/albumin concentrations." (Lines 180-181)

Regarding pronase: This is an interesting suggestion, which we have thought about before but never examined in detail. Mostly, because it is not generally used in human biomonitoring as the digestion with proteolytic enzymes (pronase, trypsin) would not result in the increase of the parent xenobiotic but in the formation of digest-specific adducts, meaning that the compound would still be bound to amino acid residues. Including those into any targeted method would require the addition of their (optimized) MRM-transitions in the method and the synthesis of authentic reference standards. Moreover, frequently these adducts only result from highly electrophilic xenobiotic metabolites and not from the parent compounds. Thus, no standardized approaches for the usage of pronase treatment for a variety of adducts and the corresponding bioinformatic data evaluation are available in the literature to the best of our knowledge. Furthermore, the assays that have been developed (aflatoxin B1-lysine adduct used for quantitating aflatoxin exposure in plasma; see e.g. McCoy et al. (2005) *Rapid Commun. Mass Spectrom.* 2005; 19: 2203–2210) typically relied on SPE clean-up steps after digestion. This is problematic in the context of our newly established method, which aims to achieve a fast and straight-forward sample preparation protocol but also to cover a wide range of structurally and chemically diverse xenobiotics, of which some are likely to be lost during SPE clean up.

Nonetheless, we performed additional experiments based on the protocol established by McCoy et al. (2005) to test the treatment proposed by the reviewer. We used pooled plasma and serum (either native or fortified and incubated with the 81 toxicants included in the method) and analysed the samples with and without pronase treatment for comparison purpose. The targeted LC-MS/MS method was adopted in a way to include theoretical adducts that might be present in fortified plasma and serum samples after digestion. This includes e.g. cysteine, lysine and tyrosine adducts. In addition, the increase in free amino acids was tracked and confirmed in treated samples to verify successful digestion (we included the corresponding MRM-transitions of the free amino acids also in our method and confirmed a strongly increased signal in comparison to untreated samples with authentic reference standards).

The objectives were:

- 1) The determination of possible covalently bound xenobiotic adducts onto albumin
 - a. Especially reactive residues on albumin are Cys34, His 146 and Tyr 411
 - b. Cross reference with reactive groups on xenobiotics in the method's standard master mix
- 2) The determination of differences in spectra prior and post digestion with pronase
 - a. Checking for proper digestion by tracking single amino acids
 - b. Checking for change in xenobiotics included in the method
 - c. Checking for prior untracked adducts

Detailed experimental design: All samples were handled in Eppendorf tubes and on ice and each sample was processed in triplicate. Aliquots (200 µL) of both commercial pooled plasma and serum (Sigma) were spiked with 20 µL of the multi-analyte xenobiotic mixture containing the 81 compounds included in the method, followed by incubation for 3 h at 37 °C in a thermoshaker, together with two processing blanks (200 µL water + 20 µL ACN). To another 200 µL aliquot of pooled plasma and serum, respectively, 20 µL ACN was added to act as a control to the fortified samples. One half of all spiked and unspiked samples, as well as blanks, were then treated with 100 µL pronase (Millipore Corp, 25 kU, from *Streptomyces griseus*) in PBS buffer (pH 7.7) at a concentration of 26 mg/mL. The other half received 100 µL PBS-Buffer without enzyme as control. All samples, including the blanks, with and without pronase treatment were incubated overnight (17 h) at 37 °C.

Afterwards, the extraction procedure was applied: 780 µL of extraction solvent (ACN and MeOH, 1:1, v/v) were added to each sample, vortexed and sonicated for 10 min in an ice bath. Precipitation of proteins was performed at -20 °C for 2 h. After centrifuging at 18.000 x g and 4 °C

for 10 min, supernatants were transferred to a new tube and the transferred volume of each sample was noted. Using a CentriVap vacuum concentrator all samples were dried overnight.

Dividing the noted volumes of the supernatants by the net volume of the respective tubes resulting from the procedure prior to drying ($V(\text{supernatant}) / 1100 \mu\text{L}$), a factor F was calculated. Each sample was reconstituted in a volume corresponding to the original sample volume (200 μL) times F with 10% ACN in water to account for lost substance due to pellet sizes at the transfer step. After centrifugation at 18.000 x g and 4 °C for 10 min, the volumes were transferred to micro inserts in LC-vials and stored at 4 °C until analysis.

The LC-MS measurement followed the same method as outlined in the manuscript with the addition of various MRM-transitions corresponding to free amino acids and hypothetical adducts. Included transitions according to thorough literature search (*selection*):

Suspected Adduct	Precursor ion	Product ions	Ionization mode
PhIP-Lysine	354.2	309.2 / 291.0	pos
PhIP-Cysteine	329.2	242.2 / 122.0	pos
PhIP-Tyrosine	389.2	344.2	pos
Naphthoquinone-Lysine	303.0	159.0 / 147.0 / 244.1	pos
Naphthoquinone-Cysteine	278.0	159.0 / 122.0 / 219.0	pos
HMF-Cysteine	248.0	127.0 / 122.0 / 109.0	pos
HMF-2xCysteine	351.0	127.0 / 122.0 / 109.0	pos
Acrylamide-Cysteine	191.0	72.0 / 122.0	pos
Glycidamide-Cysteine	209.0	90.0 / 122.0 / 44.0	pos
Dehydropyrrolizine-Cysteine	257.0	154.0 / 122.0 / 152.0	pos
Dehydropyrrolizine-2xCysteine	360.0	154.0 / 122.0 / 137.0	pos
Triclosan M1-Cysteine*	421.9	268.8 / 120.0 / 35.0	neg

* Refers to M1-metabolite as determined by Meixian, 2020
(<https://www.sciencedirect.com/science/article/pii/S0160412019336013>)

Results: The differences in signal intensity of the most abundant peak between fortified pronase treated samples, and native pronase treated samples at the theoretical MRM-transitions of selected single-residue adducts that we screened for were minimal in both matrices and within method variability. Thus, we cannot conclude that adduct-formation took place in the fortified or native samples in this *in vitro* experiment (see objective 1 above).

Figure 1: Differences in signal intensities between fortified and native samples at the theoretical MRM-transitions. Selected single-residue adducts were screened for by targeted LC-MS/MS after pronase treatment. Generally, the changes in intensity were minimal highlighting that covalent albumin binding seems not to be very abundant for most of the target analytes and that interactions were not obvious *in vitro* in our preliminary experiment.

At the same time, however, the difference in signal intensity/recovery between pronase treated and untreated samples, despite the same spiking concentrations, was found to be >20% in more than half of the analytes. This might be caused either by contamination of the pronase with the analyte in question, or by contamination with other enzymes (we suspect the purchased pronase to not be 100% pure; we have seen this with many other enzymes before) that might interact with the analytes or other sample constituents in an unpredictable manner. This suggests a negative effect of the enzymatic treatment on our method that cannot be easily compensated for; similar adverse effects of enzymatic treatment have been found in our other additional experiments using beta-glucuronidase/sulfatase (kindly see reply to referee #1; many xenobiotics can be found in the enzymes of *Helix pomatia*). From this we concluded that, with our current protocol, pronase would be likely to rather “mask” our analytes than release “hidden metabolites” in the vast majority of target toxicants (see objective 2 above).

Based on these results we concluded that pronase treatment will not be a suitable additional step in our workflow (major adoption of MRM-method needed; not feasible for high-throughput sample prep; potential contamination of samples since the pronase showed to be impure and to introduce noise, no authentic reference standards for adducts available). In addition, we noted that pronase treatment enhanced matrix effects during ionization for >60% of the analytes in serum (and similar in plasma) which in turn decreases the sensitivity of the assay (most likely due to more co-eluting molecules (impurities, breakdown products, ...) and higher competition during the electrospray process).

However, we now highlight the possibility to further examine proteolytic enzymes in future work in case the target analytes are prone to building covalent albumin adducts as follows:

“The use of proteolytic enzymes (e.g. pronase) in future work might be considered if highly electrophilic or reactive molecules with an affinity to form albumin/protein adducts are in focus.” (Lines 395-397)

Also, I could not verify how the authors use the 50 μL approach, when the method is validated for 200 μL of matrix. This is definitely not a straight-forward extrapolation, and required some in-depth research, applying these small volumes, definitely in terms of preconcentration.

The sample preparation procedure was only adjusted if less than 200 μL of plasma was available. The criticism on the extrapolation is valid; consequently, we conducted additional experiments to verify our assumption. As outlined in the revised 'Online Methods' section, we extracted 3 x 200 μL and 3 x 50 μL of pooled plasma and compared the results for a number of key analytes. The results are reported in new Table S12:

Table S12: Comparison of analyte recoveries after spiking different initial plasma volumes. Additional extraction recovery experiments to compare analyte recoveries between 50 μL and 200 μL plasma volumes were carried out in triplicates. Data evaluation was conducted for representative compounds where the commercial pooled AB plasma did not interfere with method selectivity. Differences are within the method variability determined by method validation of serum.

Compound	Spiked concentration [ng mL ⁻¹]	R _E (50 μL) / R _E (200 μL) [%]	RSD (50 μL) [%]	RSD (200 μL) [%]
Plasticizer/Plastic Components				
Bisphenol A (BPA)	1	89	5	9
Bisphenol S (BPS)	0.02	89	24	39
Phytoestrogens and Metabolites				
Daidzein	0.05	84	48	14
Enterodiol	0.05	80	3	11
Mycotoxins and Metabolites				
Alternariol	1	84	9	2
α -zearalanol (α -ZAL)	0.5	91	7	4
Zearalenone (ZEN)	0.3	100	10	5
Personal Care Product Ingredients, Pharmaceuticals and Metabolites				
Benzophenone 1	0.2	87	5	4
Butylparaben (BP)	0.1	106	14	6
Ethylparaben (EP)	0.1	138	5	3
Triclosan	1	80	15	8

R_E ... extraction recovery, RSD ... relative standard deviation

The results demonstrate that the use of a lower volume did not impact the results to any significant degree, as the variance in extraction efficiency is similar to- and mostly smaller than the validated RSD of the method. This is also in line with our abundant experience in analyzing low-volume samples in other assays and cohorts. The following information was added in the manuscript:

“Additional spiking experiments verified the proper performance of the procedure even when done with 50 μL (see Supplementary Table S12).” (Lines 179-180)

The introduction reads well but is sometimes too general, and lacks some important detailed inclusions, such as:

- L.46: 'genomic research': what do you mean?

Thank you for this comment, we elaborated on this statement and also changed some other details in the introduction.

“Complementing genomic research, the study of how genetic changes and predispositions affect gene product functions which may lead to disease, by taking the holistic environmental influences into account is key to bridging the gap between the knowledge of underlying disease mechanisms and epidemiology^{2,4}.” (Lines 35-38)

- L.55: ‘unbiased data-driven’: what do you mean?

This was written in the context of the questionnaires that are mentioned later (i.e. objective numeric data instead of self-reported intakes). We re-phrased this part to be clearer. Thank you for bringing this to our attention:

*“However, while genome-wide association studies (GWAS) were enabled by technical advancements in the past, the required tools to establish an environmental counterpart are still missing to date².”
(Lines 45-47)*

- L.64: questionnaires, specify more, not only FFQ-questionnaires, also 24h recalls, also duplicate diet studies should be mentioned here.

Thank you for this input, we now added these additional possibilities:

*“Currently available human biomonitoring (HBM) data and methods, however, clearly fall short of the need for holistic exposome research. Contrary to GWAS, the assessment of chemical exposure has historically primarily relied on self-reported questionnaires concerning food consumption and product usage¹². Particularly related to quantity, data that is dependent on personal memory (i.e. 24-hour dietary recalls) is unreliable, while duplicate diet studies are time and resource intensive.”
(Lines 53-57)*

- L.83: what is the definition of ‘ultra-trace’?

This is an interesting question and any definition will be subjective to some extent. In literature “ultra-trace” is described as a fluid concentration below 10 ng/mL according to the Clinical and Laboratory Standards Institute (CLSI, Control of Preanalytical Variation in Trace Element Determinations; Approved Guideline *CLSI document C38-A*, Wayne, PA, 1997). We reach this threshold for the majority of compounds in our method. For many analytes we actually achieved LOD values of 3-4 orders of magnitude below this value. Importantly, when the guideline was published, only single-analyte assays were used and no multi-methods with close to 100 analytes.

We appreciate the question as this is analytical jargon is used a lot without proper reflection. We now added the cited definition in the manuscript, though such notations will always remain unspecific to some extent.

*“The assessment of a multitude of organic xenobiotics at ultra-trace levels (< 10 ng mL⁻¹) in exposome research can be achieved by liquid chromatography coupled to mass spectrometry (LC-MS).”
(Lines 75-76)*

- L.99: low-volume samples: what is the definition of low volume? In my opinion, 200 µL is still ‘ok’, less than 100 µL is low volume.

We agree with the referee’s assessment and added this information in the manuscript:

“To demonstrate the potential of the method for detecting trace-level chemical contamination in precious, low-volume samples (< 100 µL), a cohort of extremely premature babies is recruited.” (Lines 92-93)

- Figure 1: the figure does not reflect what is included in the introduction (pathogenesis, precision medicine?)

The introduction was modified to include these aspects as follows:

“The correlation of comprehensive biomonitoring data at a population level with epidemiological data provides a new approach to conceptualize and unravel the contribution of chemicals to the development of disease and may enable individualized treatments for highly exposed patients in the context of precision medicine. Moreover, such information can support informed policy making to protect, in particular, susceptible demographics from adverse chemical exposures via precision prevention (Figure 1)¹⁹ to, e.g., endocrine-disrupting phthalates²⁰ and toxic perfluorinated alkylated substances (PFAS)^{21,22}.” (Lines 81-88)

The results section is well-elaborated:

- Why the choice to use the EC 657/2002 for validation, and not extend it with the ‘Bioanalytical method validation of the US Department of Health and Human Services. Guidance for industry. 2018’?

This is a topic that we discussed with other analytical chemists in Europe, the US and elsewhere abundantly in the past. There are differing opinions on which validation guideline to apply and there are many good justifications to go either way.

Here, we decided to go with the EC guideline as it is quite general and broad in scope, not too discipline/application specific, and often regarded as the gold standard in Europe. In the past we also occasionally used the FDA or EuraChem guidelines so we are aware of them and typically carefully consider which guidance to use for a specific workflow. The FDA „Bioanalytical Method Validation for Industry“ or the European counterpart („Guideline on bioanalytical method validation“ by the EMA are both designed for industrial/pharma settings relating to drug/pharmacokinetic/toxicokinetic analyses in clinical studies. The criteria set in these guidelines are typically tailored for methods containing few pharmaceutical ingredients with concentrations far above ultra-trace levels. Furthermore, there are no concentration level dependent criteria (for instance higher tolerable RSDs for very low concentrations). The EC guideline is also intended for trace level food safety applications and thus more suitable in our opinion. It also seems to be the most applied guideline in the context of human biomonitoring in Europe.

A brief literature search confirmed our assumption that hardly any research group is applying pharmaceutical industry-standards in exposome research. One simply cannot reasonably apply such guidelines made for single analytes in a highly regulated environment/industry to exploratory research containing multiple fold higher amounts of structurally highly diverse compounds. We would like to highlight that also the EC guideline is extremely thorough (and the basis for the EU legal framework e.g. in the highly regulated food safety framework) and that it was a Herculean effort by the team to evaluate all these validation parameters for such a large set of analytes.

- Not much is mentioned on the sample collection timing; the sampling of the breastmilk was done in a longitudinal way in 211 days, however what was the rationale of the 86 samples? What was the timing of the sampling: morning/evening/...? It is clear from recent papers that sample collection time is crucial, and more specifically that it is better to collect a 24h-matrix, instead of one-single-timepoint. How does this reflect (dietary) exposure?

We agree that single time point measurements are less informative than aggregate samples. In our study, samples were pooled milk obtained from multiple time points throughout a day. Given the pooled nature of our samples and the long sampling period, we believe that (dietary) exposure is reflected in a comprehensive manner. We added more details on the sample collection and timing in the *Online Methods* section:

“Spanning a period of 3-211 days post-partum, 86 pooled breast milk samples were collected from one mother and stored at -20°C until analysis. Multiple samples of pumped breast milk were collected throughout a day and partially up to 48 h with interim storage at 4°C, and combined to an aggregate sample. This was done depending on the needs of the mother and the infant, i.e. left-overs not consumed by the infant were mixed at different times of day. This approach was deemed the best option to obtain representative, pooled samples without interfering with the breast milk consumption of the infant as pointed out before by Braun et al. (2020)⁵⁷ who tested these samples for mycotoxin contamination. On some days, all breast milk was needed for feeding the infant, leaving no sample for laboratory analysis. Breast milk sampling was approved by the ethics committee of the University of Vienna (00157).” (Lines 522-531)

Could the authors make a preliminary screening on how the exposures were related to each other in a small model, besides the dynamics? As stated, this is interesting, however, only exploratory.

We thank you for this suggestion. A correlation matrix was added as Table S13. However please note the technical disclaimers describing low statistical power and the potential introduction of artificial correlations as a result of working with left-censored data.

- Why only a high-exposure scenario was applied? To work with left-censored data, I assume different scenarios should be applied to account for sensitivity analyses.

The idea behind this exposure scenario was to display the ways of how such biomonitoring data may be used by toxicologists for risk assessment. If e.g. no TDI exceedances are noted in a worst-case, high-exposure scenario, risk assessors can assume that there is no associated risk. This approach was used before (e.g. <https://doi.org/10.1016/j.chemosphere.2021.132226>) and seems to be a good first step to assess exposure/risk without doing a fully risk assessment (which would be beyond the scope of this work). Our data set clearly indicates that this upper bound scenario does not represent a threat to the infant according to current reference values, hence there was no need to create additional (maybe more realistic) data sets if the worst case has already been demonstrated not to be of concern according to current health-based reference values. We believe that expanding too much on this would distract from the manuscripts main message.

The discussion reads well with focus on limitations of the study, along with the general exposure-research field, along with general recommendations for future ExWAS.

We thank you very much for this encouraging comment.

Online methods

- The use of plastic Eppendorf tubes: did the authors take into account possible adhesion of toxicants?

We are well-aware of this possibility and therefore thoroughly evaluated this issue during the extensive analytical validation of the respective recovery efficiencies and their application to correct the generated data of unknown samples.

- Did the internal standards mixture account for all compounds? What was the rationale to use the specific concentrations of the mixture?

The applied internal standard mixture is described in the *Supplementary Information* and consisted of 13 representative analytes included in the assay. We were not able to include internal standards for all analytes since many are not commercially available and those that are can be very expensive. The applied concentrations were calculated to be around 5-10 x LOQ in the least sensitive matrix based on pre-experiments. After additional method optimization and validation, the concentration levels are now in the range of 5-20 x LOQ for most compounds which allows for proper peak integration. Since we intend to apply this workflow in large-scale cohort studies, it was important to reliably detect the IS, but also to not use too high quantities which would increase the overall cost.

This is now mentioned in the SI:

“The applied spiking concentrations were chosen to reliably detect the IS, but also to avoid wasting needlessly high quantities in order to keep the method cost-effective and thus suitable for large-scale applications.” (Lines 71-73)

- Assessing linearity: is this the most optimal solution? Lack-of-fit test, taking weighing factors into account?

We agree that this is perhaps not the perfect solution. Additionally describing the variation of the data around the calibration model as the lack-of-fit (LOF, or alternatively Mandel's Fitting) is a valid proposal. It is possible that high R^2 models still yield significant LOF. Indeed, ideally one would have both the R^2 close to 1 and non-significant LOF. If significant LOF is observed with the linear model, better fitting quadratic calibration may be applied to remove systematic biases. However, while fitting a curve to a dataset which should be expected to be linear may provide better fit, it would be difficult to justify, as in mass spectrometry the response is linearly proportional to the number of ions reaching the detector (given proper method setup). This would imply a confession that the system is not working as intended (non-linear ESI for the selected compound in the extracted matrix). Moreover, this would also have severely complicated the approximations of LODs/LOQs. The linear calibration coefficient factor R^2 has been simple, fast and therefore standard practice to assess linearity in our field so far. We are not aware of similarly comprehensive work applying these different, less straight-forward to use linearity evaluation schemes. Currently, to the best of our knowledge, there are no recommendations with regards to the circumstances which they should be applied. In this regard, it is true that, besides the technical novelty, we did not focus on the implementation of more advanced statistical evaluations of the linearity with our validation scheme. This is a topic that falls well within the discussion above dealing with the various validation guidelines and asking the questions whether they are sufficiently contemporary and appropriate for the method at hand. We highly appreciate you raising this issue.

We would like to acknowledge the valuable feedback from all three expert referees and the editor and are confident that the performed follow-up experiments and the addressed technical concerns and additional explanations/expansions greatly strengthened this work.

Reviewer comments, second round -

Reviewer #1 (Remarks to the Author):

The authors have addressed and answered most questions raised by this reviewer.

Still several corrections are needed as follows:

- Caption of figure 2 should be written as follows (numbering of subfigures):

a-b Overview of the generic sample preparation and LC-MS/MS workflow developed. c Included chemical classes. Chromatograms from scheduled multiple reaction monitoring (sMRM) show the transitions used to quantitate selected analytes in infant plasma and breast milk. Chromatograms for the corresponding matrix-matched standards are plotted in black, while the plasma (d) and breast milk (f) samples are displayed in red and blue respectively. MS2 spectra generated from enhanced product ion scans (EPIs) of reference standards are compared to the spectra acquired in biological samples for selected analytes (5-hydroxymethylfurfural, perfluorooctanesulfonic acid, jacobine-n-oxide and PhIP; technical details of EPI scans are provided in the Online Methods section). The precursor ion m/z is annotated together with fragment ions that were used for quantitative and qualitative measurements. Pie charts indicate the distribution of the detected compounds in baby plasma (e) and breast milk (g) samples according to origin or chemical classification.

- Line 18: 'ultra-sensitive'. Should be replaced by highly sensitive.
- Line 87, figure 1: This figure should be deleted. It would be more for a review paper rather such a technical paper.
- Lines 246-259 should be edited and can be shortened.
- Line 247: Replace 'expected chemical properties' by 'known chemical and biological properties'.
- Line 252: replace 'samples' by 'days'.
- Line 252: replace 'prominent' by 'striking'.
- Line 260: Why 'Moreover' ? This paragraph does not seem linked to the previous one, neither to the following paragraph.
- Line 281: 'For the first time' should be deleted. Isoflavones or equol have been known to be present in breast milk for some time. See: <https://pubmed.ncbi.nlm.nih.gov/8665689/>
- Lines 338-340: Sentence not clear. Rewriting needed.
- Line 291: replace 'merely' by 'only'.
- Line 392: replace 'Moreover' by 'In addition'.
- Lines 393-399: Not easily read. Rewriting needed.
- Line 397, 'increased toxicant concentrations': Not always true according to table S11. In particular, some results are unexpected. For example, concentrations in breast milk are significantly higher after enzyme treatment (expected), but the number of samples where detected very much decreases after treatment (I guess this is what 'pos. samples' means; abbreviation should be developed in a note of the table). This is unexpected and in contradiction with the increase of concentrations after treatment.
- There might also be some errors in the concentrations of 8-prenylnaringenin after enzymatic treatment (this time the number of samples where detected increases, but the concentrations remain unchanged).
- Line 408: 'Excretory' ?
- Lines 408-410: These lines are not needed here.
- Lines 413-415: This sentence is not needed.
- Lines 418-420: Why would phase-II metabolites solve the problem? They would only capture acute exposures just as parent compounds.
- Lines 420-428: These lines deal with biomarkers of effect. They are out of place at this point of the discussion.
- Lines 428-430: 'mechanistically simpler' ?? Adducts would require a dedicated discussion in a separate paragraph, with some citations of recent papers on the adductome.

Reviewer #2 (Remarks to the Author):

The authors have provided an outstanding and thoughtful response to all of the reviewer comments. The manuscript is stronger and the message is clear. I have no further concerns.

Reviewer #3 (Remarks to the Author):

Based on the extensive authors rebuttal, I am highly confident of the quality of this innovative paper. Additionally, I appreciate the extra experimental efforts made to elucidate my concerns regarding the used methodology. Congratulations!

Reviewer #1 (Remarks to the Author):

Still several corrections are needed as follows:

- Caption of figure 2 should be written as follows (numbering of subfigures):

a-b Overview of the generic sample preparation and LC-MS/MS workflow developed. c Included chemical classes. Chromatograms from scheduled multiple reaction monitoring (sMRM) show the transitions used to quantitate selected analytes in infant plasma and breast milk. Chromatograms for the corresponding matrix-matched standards are plotted in black, while the plasma (d) and breast milk (f) samples are displayed in red and blue respectively. MS2 spectra generated from enhanced product ion scans (EPIs) of reference standards are compared to the spectra acquired in biological samples for selected analytes (5-hydroxymethylfurfural, perfluorooctanesulfonic acid, jacobine-n-oxide and PhIP; technical details of EPI scans are provided in the Online Methods section). The precursor ion m/z is annotated together with fragment ions that were used for quantitative and qualitative measurements. Pie charts indicate the distribution of the detected compounds in baby plasma (e) and breast milk (g) samples according to origin or chemical classification.

We would like to thank the reviewer for the time invested to re-evaluate our revised manuscript in great detail and the suggestions for further improvements. We adapted the caption of Figure 2 as recommended by the referee.

- Line 18: 'ultra-sensitive'. Should be replaced by highly sensitive. **Done.**

- Line 87, figure 1: This figure should be deleted. It would be more for a review paper rather such a technical paper.

This point was discussed with the editor before who agreed to keep this figure as it has value for the broader readership of *Nature Communications*.

- Lines 246-259 should be edited and can be shortened.

- Line 247: Replace 'expected chemical properties' by 'known chemical and biological properties'

- Line 252: replace 'samples' by 'days'.

- Line 252: replace 'prominent' by 'striking'.

We appreciate the suggestions, which all were included in the revised version of the paragraph.

- Line 260: Why 'Moreover' ? This paragraph does not seem linked to the previous one, neither to the following paragraph.

We agree and re-phrased the sentence accordingly: "As a tool to reveal potential trends and associations between chemical exposures, a correlation matrix is shown in Supplementary Table S13."

- Line 281: 'For the first time' should be deleted. Isoflavones or equol have been known to be present in breast milk for some time. See: <https://pubmed.ncbi.nlm.nih.gov/8665689/>

This statement does not refer to the isoflavones or equol but to the phytotoxins (pyrrolizidine and tropane alkaloids) which indeed were detected for the first time. We changed the wording to enhance clarity.

- Lines 338-340: Sentence not clear. Rewriting needed.

Thank you for the comment. We re-wrote this sentence as follows: *“Alongside the assessment of method linearity, sensitivity and selectivity, two trace concentration levels were applied in spiking experiments to determine analyte recoveries. The method yielded highly convincing results during validation for urine and serum, resulting in more than 75% of the chemicals included fulfilling all criteria for at least one of the fortification levels.”*

- Line 291: replace ‘merely’ by ‘only’. **Done (Remark: we assume this comment refers to line 391)**

- Line 392: replace ‘Moreover’ by ‘In addition’. **Done**

- Lines 393-399: Not easily read. Rewriting needed. **We adapted to make this part easier to understand for the non-analytical reader.**

- Line 397, ‘increased toxicant concentrations’: Not always true according to table S11. In particular, some results are unexpected. For example, concentrations in breast milk are significantly higher after enzyme treatment (expected), but the number of samples where detected very much decreases after treatment (I guess this is what 'pos. samples' means; abbreviation should be developed in a note of the table). This is unexpected and in contradiction with the increase of concentrations after treatment.

Thank you for this comment. The abbreviation “pos. samples” is now explained in the table legend as requested. The reason why some of the numbers for positive samples were decreased was explained in the following comment during the first revision of the paper:

‘The additional experiments confirmed our assumption that for many toxicants increased concentrations can be found (e.g. most phytoestrogens). However, as also mentioned by the referee, applying this treatment is not always straight-forward. The reasons were as follows: (1) Contamination of the crude snail extract with the xenobiotic and thus a high background signal, (2) retention time shifts out of the sMRM window caused by the pH of the utilized buffer, (3) a general higher background noise in samples, which we could also see in other assays after enzymatic treatment in the past, that reduces the method’s sensitivity. Consequently, some analytes could not be assessed at all after treatment in breast milk (e.g. phytotoxins and PhIP).’

Also, matrix effects might be modulated by the enzymes. Consequently, we do not believe that this finding is unexpected for some compounds when considering the technical issues potentially introduced by the enzymatic treatment.

- There might also be some errors in the concentrations of 8-prenylnaringenin after enzymatic treatment (this time the number of samples where detected increases, but the concentrations remain unchanged).

It is correct that the number of positive identifications is increased. In the reported table only the min and max concentrations are reported and the max concentration was slightly increased. We cross-checked the data and can confirm its validity.

- Line 408: ‘Excretory’ ? **Exchanged with “excreted”. Thank you.**

- Lines 408-410: These lines are not needed here.

We thank for the comment. We understand the remark that this is not needed, but the reference to the possibility of colorimetric assays was requested by reviewer #3. His/her request was also the

reason why the clinical data on albumin/hematocrit of the neonate cohort was included during the revision process. Therefore, we would prefer to keep these lines.

- Lines 413-415: This sentence is not needed. **We agree and removed it.**

- Lines 418-420: Why would phase-II metabolites solve the problem? They would only capture acute exposures just as parent compounds.

We fully agree; this is why we mention that *longitudinal* analysis would be needed in such a case where no long-lived metabolites are available/known. The sentence was adapted to better transport this message and increased clarity.

- Lines 420-428: These lines deal with biomarkers of effect. They are out of place at this point of the discussion.

The discussion of biological effect monitoring vs exposure monitoring was again an important aspect raised by reviewer #3. His/her point was: “... *however the question is ‘are we looking to the right molecules’? I miss this section in the manuscript, are the compounds included the real validated biomarkers of exposure and/or effect?’*” For this reason, we included this additional paragraph and feel that this adds significantly to the overall discussion of the manuscript. We would prefer to keep this part of the discussion.

- Lines 428-430: ‘mechanistically simpler’ ?? Adducts would require a dedicated discussion in a separate paragraph, with some citations of recent papers on the adductome.

We agree and removed the attribute “*mechanistically simpler*” accordingly.

We would like to acknowledge the valuable additional feedback and are confident that the remaining questions were fully addressed.

Documentation Figure 1:

This figure was created using the open source graphics editor inkscape (<https://inkscape.org/>). The included icon library of scalable vector graphics (svg) can therefore be used and distributed freely.

Additional sources of svgs and pictures included and/or modified for Figure 1 were:

<https://reactome.org/icon-lib>
<https://www.svgrepo.com/>
<https://freesvg.org/>
<https://svgsilh.com/>
<https://www.pngkey.com/>

All of these providers are of open source/public domain utilizing the creative commons public domain license (<https://creativecommons.org/publicdomain/zero/1.0/>), therefore allowing the copying, modification and distribution (even for commercial use) of graphics without asking for additional permission.